

# Yttrium speciation in subduction zone fluids from *ab initio* molecular dynamics simulations

Johannes Stefanski and Sandro Jahn

Institute of Geology and Mineralogy, University of Cologne, Zülpicher Str. 49b, 50674 Cologne, Germany

**Correspondence:** Sandro Jahn (s.jahn@uni-koeln.de)

**Abstract.** The rare earth elements (REE) are important geochemical tracers for geological processes such as high grade metamorphism. Aqueous fluids are considered important carriers for the REE in a variety of geological environments including settings associated with subduction zones. The capacity of a fluid to mobilize REE strongly depends on its chemical composition and on the presence of suitable ligands such as fluoride and chloride. In this study we present structural and thermodynamic properties of aqueous yttrium chloride and fluoride species at a temperature of $800\,^{\circ}\mathrm{C}$ in a pressure range between 1.3 and $4.5\,\mathrm{GPa}$ derived from *ab initio* molecular dynamics simulations. The total yttrium coordination by $H_2O$ and halide ions changes from seven to eight within the pressure range. For the yttrium chloride species a maximum number of three chloride ligands was observed. The derived thermodynamic data show that aqueous yttrium fluoride complexes are more stable than their yttrium chloride counterparts at conditions relevant to slab dehydration. Mixed Y-(Cl,F) complexes are found to be unstable. Furthermore, in contrast to field observations thermodynamic modeling indicates that yttrium should be mobilized at rather low fluoride concentrations in high-grade metasomatic systems. These results suggest a rather low fluoride activity in the majority of subduction zone fluids because yttrium is one of the least mobile REE. Additionally, the simulations indicate that yttrium drives the self-ionization of hydration water molecules as it was observed for other high field strength elements. This might be a general property for highly charged cations in aqueous solutions under high temperature and high pressure conditions.

## 1 Introduction

Subduction zones have been the most important sites for exchange of matter and energy between the Earth's crust and mantle for billions of years until now (Tang et al., 2016). Magnetotelluric anomalies (e.g. Worzewski et al. (2011); McGary et al. (2014)) suggest the occurrence of a high proportion of melts and water-rich fluids in the subducted slabs due to partial melting (Zheng et al., 2016) and due to the dehydration of water-bearing minerals such as serpentine (Ulmer and Trommsdorff, 1995) and amphibole (Schmidt and Poli, 1998). Quite naturally these fluids do not consist of pure water much rather they are brines



with high salinity up to several mass percent of Cl (Métrich and Wallace, 2008; Newton and Manning, 2010), and contain Si, Al, and alkali cations (Na, K) as major solutes with minor amounts of Ca, Fe, and Mg (Manning, 2004; Hermann et al., 2013).

Aqueous fluids play an important role in subduction zones. Their interaction with minerals and rocks result in alterations including the dissolution and precipitation of minerals and/or the exchange of chemical elements and isotopes. Furthermore, fluid-mediated transport of trace elements such as high-field-strength elements (HFSE) and rare earth elements (REE) is a major process of the deep element cycles (Manning, 2004). The distribution of these elements between minerals and fluids or melts is used as petrogenetic indicator for fractionation processes in igneous and metasomatic petrology (Winter, 2009). It is

known that REE patterns of subducted rocks are affected by the chemical composition of the metamorphic fluid (e.g. John et al. (2008); Zhang et al. (2008)) due to their chemical complexation with dissolved anions, e.g. $F^-$, $SO_4^{2-}$, $CO_3^{2-}$ and $Cl^-$ (Tsay et al., 2014, 2017; Alt et al., 1993; Scambelluri and Philippot, 2001; Newton and Manning, 2010).

The speciation of REE at moderate pressures (up to few 100 MPa) and high temperatures (250-300 °C) was studied experimentally using *in-situ* X-ray absorption spectroscopy (XAS) and solubility experiments (see recent review by Migdisov et al.

(2016)) to understand physicochemical properties of hydrothermal fluids related to REE ore deposition focusing on chloride and fluoride complexes. Due to the high stability of fluoride complexes (Wood, 1990; Haas et al., 1995) it is a widely shared notion that fluoride complexes are most important for REE transport in hydrothermal fluids but Migdisov and Williams-Jones (2014) suggested that REE fluoride complexes are not the major carrier of REEs due to the low solubility of REE fluoride minerals such as bastnaesite, $(Ce, La, Nd, Y)[FCO_3]$, and due to the low fluoride activity in low $pH$ environments. According

to Migdisov and Williams-Jones (2014), REE chloride and sulfate complexes appear to be the main species for REE transport in hydrothermal systems. However, this interpretation is questioned by other authors (Xing et al., 2018).

So far, the number of *in situ* studies that address the complexation or thermodynamic properties of REE aqueous species at pressure ($P$) and temperature ($T$) conditions of subduction zones is very limited due to the challenging experimental setups (Sanchez-Valle, 2013). Apart from field observations (e.g. fluid inclusion analysis), our main understanding of the behavior of

REE under high $P/T$ conditions is derived from fluid/mineral partitioning and solubility experiments (e.g. Bali et al. (2012); Keppler (1996); Tsay et al. (2014); Keppler (2017)) and from numerical simulations. In a case study, van Sijl et al. (2009) modeled the hydration shell of REEs in solution by static energy calculations of an explicit first hydration shell and an implicit solvent model. Temperature effects were introduced by considering changes of the dielectric constant of the solvent and by calculations of the entropy. In this study it is concluded that the hydration energies of all lanthanides become more similar at

high $P/T$ conditions and that the availability of ligands becomes a controlling factor for the fractionation of light REE (LREE) and heavy REE (HREE) by subduction zone fluids. Experiments suggest that LREEs (e.g. La) are more mobile than HREE in chloride-rich solutions (Tropper et al., 2011; Tsay et al., 2014). The presence of fluoride in the system enhances the mobility of HREE and this leads to fractionation processes (Tropper et al., 2013).

From a geochemical perspective, yttrium is considered a HREE[1] and as such it is very common to use yttrium as a repre-

sentative of the whole group of HREEs because of their similar chemical properties. Further, comparable behavior of Y and the majority of the HREE in high-grade metasomatism processes (Ague, 2017) support this assumption. The hydration shell

---

[1]International Union of Pure and Applied Chemistry: Nomenclature of Inorganic Recommendations 2005





of $Y^{3+}$ in aqueous solutions and possible complexation of $Y^{3+}$ with chloride has been subject to a number of studies at room temperature (e.g. Johansson and Wakita (1985); Petrović et al. (2016)). Molecular simulations in conjunction with advanced sampling methods indicate that in the absence of other ligands $Y^{3+}$ is coordinated by eight hydration water molecules (Ikeda

et al., 2005a, b), whereas at high $pH$ $[Y(OH)_3(H_2O)_3]_{aq}$ complexes are formed (Liu et al., 2012). The reported Y-O distance of $2.38\,\text{Å}$ (Ikeda et al., 2005b) agrees with EXAFS and XANES measurements (Näslund et al., 2000; Lindqvist-Reis et al., 2000).

The complexation of $Y^{3+}$ with $Cl^-$ was studied by Vala Ragnarsdottir et al. (1998) at hydrothermal conditions up to $340\,^\circ C$ using *in situ* EXAFS spectroscopy. The authors claim that yttrium is coordinated by 8-9 neighbors at hydrothermal conditions

and does not associate with chloride but rather forms polyatomic yttrium species. In another study by Mayanovic et al. (2002), a strong association of Y with chloride up to $YCl_4^-$ at $500\,^\circ C$ and a linear reduction of the total number of coordinating atoms from eight to four towards high temperatures are reported. The results of that study indicate that yttrium behaves like 3d transition metal ions rather than a HREE under hydrothermal conditions at high chloride activity. Solubility experiments up to $1\,\text{GPa}$ and $800\,^\circ C$ in a hydrothermal piston-cylinder apparatus performed by Tropper et al. (2013, 2011) indicate that yttrium

is transported as $[YClOH]^-$ complexes in $NaCl$ brines and that $YF_2^+$ is the major complex in a fluorine-rich environment.

While the stability and distribution of Y-(Cl,F) species in aqueous solutions at ambient conditions has been subject to a number of studies (e.g. Luo and Byrne (2001, 2000, 2007) the knowledge of thermodynamic properties of yttrium species in hydrothermal fluids is limited to theoretical predictions (Haas et al., 1995; Wood, 1990) based on regressions using the Helgeson-Kirkham-Flowers (HKF) model (Helgeson et al., 1981) and one experimental study by Loges et al. (2013). Stability

constants of Y-(F,Cl) complexes at sub-crustal high $P/T$ conditions have been barely investigated.

The capacity of a fluid to mobilize a certain element or to dissolve a certain amount of a component in the fluid depends on the chemical potential of the formed aqueous complexes (Anderson, 2009; Dolejš, 2013). For a better understanding of the mobility of Y (as a representative of the HREE group) in subduction zone fluids at high $P/T$ conditions, knowledge of the relation between the concentration of the molecular species in aqueous fluids and their thermodynamic properties are required. Yttrium

in general is one of the most immobile REEs (Ague, 2017; Schmidt et al., 2007b) in high grade metasomatic environments. But the high mobility in certain locations not only associated with hydrothermal ore deposits (McPhie et al., 2011; Graupner et al., 1999) but also in metamorphic or diagenetic context (Hole et al., 1992; Moore et al., 2013; Harlov et al., 2006) indicates that the dissolution or transport of Y is constrained to a very narrow range of fluid compositions. Therefore, yttrium could be a potential indicator for certain geological fluid compositions. In this study we use *ab initio* molecular dynamics (AIMD)

simulations to investigate the atomic-scale structure and probe the free energy of different Y-(Cl,F) complexes in the $P/T$ range of subduction zones.



## 2 Methods

### 2.1 Ab initio molecular dynamics

The AIMD simulation approach is based on a quantum-mechanical description of the electronic structure within the density
functional theory (DFT) (Hohenberg and Kohn, 1964; Kohn and Sham, 1965). Here, we used AIMD simulations with the Car-
Parrinello (Car and Parrinello, 1985) method to model the molecular structure of Y-(Cl,F) complexes in aqueous solutions. We
performed simulations with the widely used CPMD code (CPMD, 1990; Marx and Hutter, 2000). Within the code the BLYP
exchange correlation functional (Becke, 1988) was employed and the plane-wave expansion of the Kohn-Sham orbitals was
truncated at a cutoff energy of $80\,\mathrm{Ry}$. To reduce the computation afford the core electrons of all atoms in the cubic simulation
cell were approximated by Goedecker-type pseudopotentials (Goedecker et al., 1996; Hartwigsen et al., 1998; Krack, 2005).
To separate the electronic and nuclear motion of the Car-Parrinello molecular dynamics, a fictive electron mass of $600\,\mathrm{a.u.}$ with
fictitious kinetic energy of $0.24\,\mathrm{a.u}$ was used. All results presented here are based on simulations performed with a constant
number of atoms $N$ at constant volume $V$ and a temperature $T$ of $800\,^{\circ}\mathrm{C}$ (so called $NVT$ ensemble) and with a time step of
$0.1\,\mathrm{fs}$. The temperature in the simulation was controlled using a Nosé thermostat (Nosé, 1984; Hoover, 1985). We chose two
different pressure conditions of $1.3\,\mathrm{GPa}$ and $4.5\,\mathrm{GPa}$ for this study. The volumes of the simulation cell were estimated from
the correlation function provided by Mantegazzi et al. (2013) assuming $2\,\mathrm{molal}\,\mathrm{NaCl}$ solution for all cells (see Tab. 1). Due to
this approximation the pressures listed in Tab. 1 have to be considered estimates.

    The initial atomic configurations were derived from AIMD simulations of pure $\mathrm{NaCl}$ solutions ($200\,\mathrm{H_2O}$ and $10\,\mathrm{NaCl}$).
This configuration had been equilibrated for a few tens of picoseconds (ps) at $1000\,\mathrm{K}$ and a simulation box size of $13.74\,\mathrm{Å}$.
The original $\mathrm{NaCl}$ solutions were generated in classical molecular dynamics simulations using MCY potentials (Matsuoka
et al., 1976). We substituted one of the sodium atoms by yttrium and decreased the number of water molecules and chlorine
atoms stepwise until we reached configuration A1 in Tab. 1. Figure 1 shows a snapshot of the simulation cell A1 with a
$[\mathrm{YCl_3(H_2O)_4}]_{\mathrm{aq}}$ complex. The chloride ions not initially bonded to the yttrium ion are constrained to remain at larger distance
($6\text{-}7\,\mathrm{Å}$) from the yttrium ion. All other simulation boxes (see Tab. 1) were generated from this initial one and equilibrated for
several picoseconds.

    For the fluoride-bearing cells we used different cell compositions due to the strong association of hydrogen and fluoride
at low pressures. Only initially bonded $\mathrm{F^-}$ ions were included to avoid the formation of hydrofluoric acid by the reaction
$\mathrm{H_2O + F^- \rightleftharpoons OH^- + HF}$.

### 2.2 Analysis of interaction distances and coordination number

The average atomic structure of disordered systems such as aqueous solutions is commonly described in terms of partial radial
distribution functions $g_{ij}(r)$. These functions describe the probability for finding a pair of atoms of elements $j$ and $i$ at a





**Table 1.** Number of atoms in the different simulation cells together with the size of the simulation cell. A and B refer to the system density $^\top$ of $1072\,\mathrm{kg\,m^{-3}}$ (1.3 GPa) and $1447\,\mathrm{kg\,m^{-3}}$ (4.5 GPa).

| cell | $H_2O$ | $Y^{3+}$ | $Cl^-$ | $F^-$ | $Na^+$ | no. atoms | $a^\dagger$ |
|------|------|------|------|------|------|-----------|------|
| A1 | 84 | 1 | 6 | 0 | 3 | 262 | 14.29 |
| A2 | 84 | 1 | 5 | 1 | 3 | 262 | 14.25 |
| A3 | 84 | 1 | 4 | 2 | 3 | 262 | 14.21 |
| A4 | 84 | 1 | 3 | 3 | 3 | 262 | 14.16 |
| B1 | 84 | 1 | 6 | 0 | 3 | 262 | 12.93 |
| B2 | 84 | 1 | 3 | 3 | 3 | 262 | 12.82 |

$\top$ volume estimated using the empirical equation of state from Mantegazzi et al.
(2013) for $2\,\mathrm{molal}$ NaCl solution, $^\dagger$ edge length of the simulation box (Å)

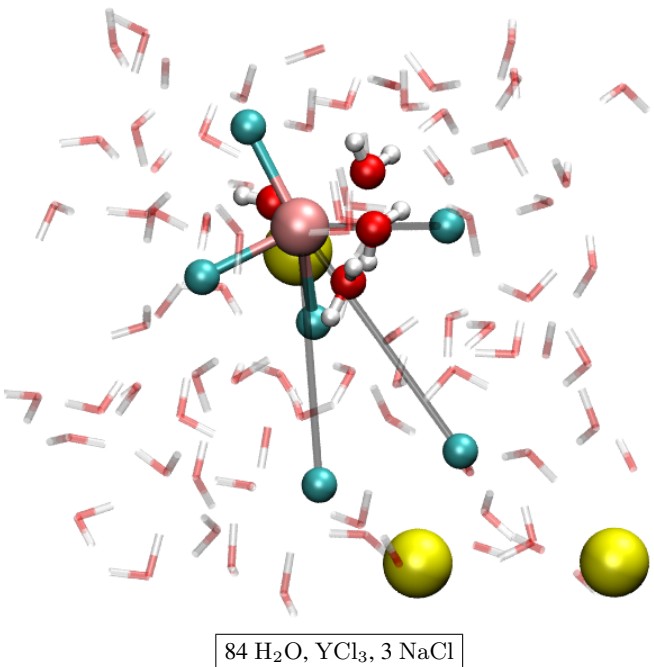

$H_2O$, $YCl_3$, 3 NaCl

**Figure 1.** Snapshot of the simulation cell A1 with a $[YCl_3(H_2O)_4]_{aq}$ complex. The water molecules are indicated by red-white bond sticks, sodium by yellow balls and chlorine by cyan balls. The $H_2O$ in first hydration shell of the yttrium atom (copper colored) are presented as red-white balls and sticks. The constraint distances between the yttrium ion and the constraint $Cl^-$ are colored in gray.

distance $r$ normalized to the particle number density $\rho_N$ of the fluid:

$$g_{ij}(r) = \frac{1}{c_i c_j 4\pi r^2 \rho_N N} \sum_{a=1}^{N_i} \sum_{b=1}^{N_j} \delta(r - |\mathbf{R}_a - \mathbf{R}_b|) \tag{1}$$





where $c_i$ $(= N_i/N)$ and $c_j$ are the concentrations of elements $i$ and $j$, $N$ is the number of particles in the simulation box, $\delta(x)$
the Dirac delta function, and $\boldsymbol{R}_a$ and $\boldsymbol{R}_b$ are the position vectors of particles $a$ and $b$. In the numerical implementation of Eq. 1
the distance of each particle in respect to all other particles is calculated at every time step. The evolving list of distances is
normalized to the number of particles and the volume of the simulation cell.

The positions of the first maximum of $g_{ij}(r)$ represent the distances with the highest density of particles around the central
position, which are usually interpreted as the nearest neighbor distances (or bond distances) between elements $i$ and $j$. The
average coordination number of one element is derived from counting the number of neighbors for each atom of this element
within a given cutoff distance respecting periodic boundary conditions and averaging over all particles of the same kind and
over time. The cutoff distance is taken from the first minimum of the respective $g_{ij}(r)$. The association of OH$^-$ groups with
cations is evaluated by considering oxygen atoms coordinated by one hydrogen only. To distinguish pure OH$^-$ from two H$_2$O
sharing one hydrogen the cutoff between the oxygen and the hydrogen is set to 1.3 Å. Additionally, the distance of the hydrogen
within this cutoff distance to the next oxygen is taken as distinguishing criterion. Only if the oxygen of the next water molecule
is located in a distance of $> 1.6$ Å the OH$^-$ is accounted as hydroxide. This value represents approximately the hydrogen bond
distance between OH$^-$ and H$_2$O (Stefanski et al., 2018). To evaluate the formation of a certain species during a simulation run
only complexes with a constant coordination of chloride and fluoride over at least 3 ps are considered. The average halogen ion
hydration number is computed by counting the number of hydrogen oriented towards the ion of the vicinal water molecules.

## 2.3 Constraint molecular dynamics simulations and thermodynamic integration

A single MD simulation only yields the total internal energy of the system. Thermodynamic integration (TI) is used to derive
free energy differences between different states. This approach usually requires a number of intermediate MD simulations
along a certain integration variable (Resat and Mezei, 1993). Here, we use the constrained molecular dynamics approach and
thermodynamic integration in terms of the blue moon sampling (Ciccotti et al., 2005). This method has been used already
by different groups to investigate the stability of metal complexes in aqueous solutions, not only at ambient pressure and
temperature conditions (Bühl and Golubnychiy, 2007; Bühl and Grenthe, 2011) but also at hydrothermal (Mei et al., 2013,
2015, 2016) and deep crustal high density fluid conditions (Mei et al., 2018).

Using this method, the Helmholtz free energy difference ($\Delta_r A$) of a chemical dissociation reaction is obtained from the
average forces $F(r)$ between two atoms under the constraint of keeping their bond distance $r$ constant. $\Delta_r A$ is derived from
the integration of $F(r)$ between two different distances $r_1$ and $r_2$, which correspond to the associated and the dissociated state
of the aqueous complexes (Sprik and Ciccotti, 1998):

$$\Delta_r A_{1 \longrightarrow 2} = -\int_1^2 \langle F(r) \rangle dr \tag{2}$$

The formal relation between the Helmholtz free energy and the Gibbs free energy is given by:

$$\Delta_r G = \Delta_r A_{1 \longrightarrow 2} + V \int_1^2 dP \tag{3}$$





$V$ is the volume of the simulation cell. Here we use an $NVT$ ensemble and the change in pressure averaged over the whole trajectory is approximately zero ($\int_1^2 dP = 0$).

To look into the formation of Y-(Cl,F) species in equilibrium reactions we removed one of the ligands from the Y ion in multiple integration steps by constraining the Y-(Cl/F) distance. During the integration the Y-Cl/F distances of the Cl or F ions that are not initially bonded to the yttrium ion are fixed at 6.0 to 7.0 Å to avoid disruptions during the integration. Figure 2

illustrates an example of the dissociation reaction in the simulation box at 1.3 GPa and 800 °C:

$$[YCl_3(H_2O)_4]_{aq} + H_2O \rightarrow [YCl_2(H_2O)_5]^+ + Cl^- \tag{R1}$$

The integration starts at the first distance (Fig. 2 (a)), which corresponds approximately to the equilibrium distance of the Y-Cl contact ion pair. This interatomic distance (a) is estimated as the intercept with the zero force line from a linear interpolation between the first and the second integration step, the first step starting at 2.6 Å for Y-Cl (and at 2.0 Å for Y-F). With increasing

displacement of the chloride ion, the constraint force is attractive (Fig. 2 (b)) until a water molecule takes the place of the ion in the first coordination shell around the $Y^{3+}$ (Fig. 2 (c)). At this point the force becomes repulsive. By integration over the potential of mean force (PMF) (Fig. 2 (I)) the Helmholtz free energy ($\Delta_r A$) difference between the initial complex and the product of the reaction is derived in (Fig. 2 (II)). For the Y-Cl complexes, we assume a ligand in a distance of 6.0 Å as being dissociated and for Y-F this distance reduces to 5.0 Å. In Fig. 2 (d) the dissociation is completed and $[YCl_2(H_2O)_5]^+$ is formed.

To estimate the convergence of the constraint force the standard deviation of the average force is computed. As convergence criterion a value of $5 \, \mathrm{kJ \, mol^{-1}}$ over the last 2 ps is taken. This value is also considered as approximate error of the computed reaction free energies. To satisfy this criterion the constraint AIMD simulations are performed for between 4.5 and 40 ps.

## 2.4   Thermodynamic approaches

It is textbook knowledge that the formation of monomeric complexes in equilibrium reactions develops in steps (Atkins and

De Paula, J., 2006; Brown and Ekberg, 2016). During the formation process a ligand L is added to the metal cation M. This formation of a $ML_n$ complex can be written as a sequence of stepwise reactions:

$$
\begin{aligned}
\mathrm{M + L} \;&=\; \mathrm{ML} \\
\mathrm{ML + L} \;&=\; \mathrm{ML_2} \\
&\;\;\vdots
\end{aligned}
$$

$$\mathrm{ML_{n-1} + L} \;=\; \mathrm{ML_n}$$

with the respective logarithmic equilibrium constants

$$log \, K_n = \frac{-\Delta_r G_n^\circ}{2.303 \, RT} \tag{4}$$

Having determined equilibrium constants for all of those reactions, the stability constant (also referred to as cumulative stability constant or overall stability constant) $\beta_n$ of species $ML_n$ is defined as

$$\log \beta_n = \log K_1 + \log K_2 + \log K_3 \cdots \log K_n \tag{5}$$





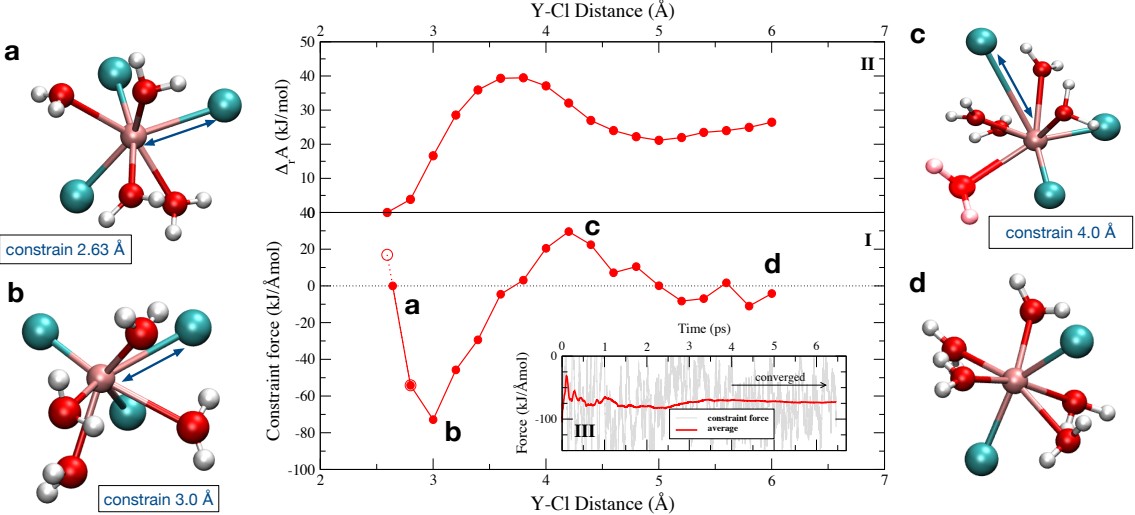

**Figure 2.** (I) potential of mean force of the dissociation reaction of $YCl_3$ to $YCl_2^+$ at $1.3\,GPa$ and $800\,°C$ over a distance between $2.63\,Å$ and $6.0\,Å$. The evolution of the Helmholtz free energy is shown in (II). (a-d) indicate the different stages of the dissociation of the initial complex (see text for details). (III) shows exemplary the progress of the constraint force with simulation time for stage b for a Y-Cl distance of $3.0\,Å$.

The standard Gibbs free energy ($\Delta_r G_i^\circ$) depends on the reaction Gibbs free energy ($\Delta_R G_i$) derived from the MD simulation, temperature $T$, gas constant $R$, molality of the ions $m_i$ and the activity coefficient $\gamma_i$:

$$\Delta_r G_i^\circ = \Delta_r G_i - RT \, \ln \frac{m_{ML_i}\gamma_{ML_i}}{m_{ML_{i-1}}\gamma_{ML_{i-1}} \cdot m_L \gamma_L} \tag{6}$$

The standard state for a solute in aqueous solution is a $1\,molal$ hypothetical solution with properties of a infinitely diluted
solution (IUPAC, 1982). The concentration and behavior of the solutes in the simulation cell (see Tab. 1) is quite different from this standard state. Therefore, we computed the activity coefficient corresponding to this hypothetical solution using the B-dot model (Helgeson et al., 1981; Helgeson, 1969), which is an empirical extension of the Debye–Hückel theory (Hückel and Debye, 1923):

$$\log \gamma_i = -\frac{z_i^2 A_{DH}\sqrt{I}}{1 + \mathring{a}_i B_{DH}\sqrt{I}} + \dot{B}I \tag{7}$$

where $z_i$ is the charge of ion $i$, $\mathring{a}$ the mean distance of closest approach between the ions and $I$ the ionic strength:

$$I = 0.5 \sum_{i=1}^{n} m_i z_i^2 \tag{8}$$





$A_{\text{DH}}$ and $B_{\text{DH}}$ are the Debye-Hückel parameters

$$A_{\text{DH}} = \frac{1.8248 \cdot 10^6 \sqrt{\rho_{H_2O}}}{(T\epsilon)^{\frac{3}{2}}} \tag{9}$$

$$B_{\text{DH}} = \frac{50.292 \sqrt{\rho_{H_2O}}}{(T\epsilon)^{\frac{1}{2}}} \tag{10}$$

They depend on temperature, density ($\rho_{\text{H}_2\text{O}}$) and dielectric constant ($\epsilon$) of the solvent. $\epsilon$ is computed using the equation provided by Pan et al. (2013) and Sverjensky et al. (2014) for pure $H_2O$ assuming a fluid density of the simulation. The value of $\dot{B}$ is calculated by the CHNOSZ software package (Dick, 2008) applying the extrapolation suggested by Manning et al. (2013). The $\mathring{a}$ parameter for the different complexes and ions (for all Y-Cl/F complexes a constant value of 4.5 Å is applied) are taken from Kielland (1937) and Eq. 7 was solved in a Python implementation of the EQBRM program (Anderson and Crerar,

1993).

Note that in the simulations we modeled a dissociation reaction but to be in line with the modern nomenclature of aqueous geochemistry all the derived thermodynamic data presented below correspond to the formation reaction. In this study, we investigated two kinds of reactions:

$$Y^{3+} + nCl^- \rightarrow YCl_n^{3-n} \text{(n=1–3)} \tag{R2}$$

and

$$Y^{3+} + nF^- \rightarrow YF_n^{3-n} \text{(n=1–3)} \tag{R3}$$

In the following all results referring to one of those reactions are indexed by TI-n. Furthermore, for reasons of clarity we do not include the number of hydration water molecules in the formula of the aqueous complexes in some of the presented figures and tables.

## 3  Results

### 3.1  Yttrium coordination in high density aqueous fluid

AIMD simulations were performed for the hydrated $Y^{3+}$ and for eleven different yttrium-halogen complexes: five $YCl_n^{3-n}$, n=1-5, three $YF_n^{3-n}$, n=1,2,3 and three mixed Y-(Cl,F) complexes. Simulation conditions and obtained structural data are compiled in Tab. 2. Moreover, the formed aqueous species are listed in Tab. 3. Note that the composition of the simulation cells

varies slightly as cells A1-A4, B1 and B2 contain different amounts of F and Cl.

Different partial radial distribution functions for Y-($Cl^-$, $F^-$, $OH^-$, $OH_2$) are shown in Fig. 3. To facilitate the comparison of the first peaks the $g_{ij}(r)$ are scaled to equal maximum intensity. The observed sequence of atomic distances between the central metal ion and its ligands holds for all the complexes. The shortest distance is found between $Y^{3+}$ and $F^-$ followed by Y-$OH^-$ and Y-$OH_2$. The largest distance is observed for Y-Cl pairs. Further, in $g_{\text{Y-O}}(r)$ a second maximum is observed.





**Table 2.** The listed atomic distances and coordination numbers are averaged over the whole AIMD time for all unbiased runs. The lifetime refers to persistence of the initial yttrium halide complex.

| | Simulation parameters | | | Distances (Å) | | | | Coordination numbers | | | | | | Lifetime (ps) |
|---|---|---|---|---|---|---|---|---|---|---|---|---|---|---|
| Run ID | $\rho$ (kg m$^{-3}$) | Time (ps) | Cell | Y-O | Y-O (2$^{nd}$) | Y-Cl | Y-F | Y-O | Y-OH$^-$ | Y-Cl | Y-F | Y(Cl,F)-Na | Y-(O,Cl,F) | |
| #1 | 1072 | 25 | A1 | 2.35 | 4.7 | 2.58 | - | 5.9 | 0.6 | 1.0 | - | 0.1 | 6.9 | 25 |
| #2 | 1072 | 23 | A1 | 2.36 | 4.8 | 2.59 | - | 4.8 | 0.3 | 2.0 | - | 0.5 | 6.8 | 23 |
| #3 | 1072 | 24 | A1 | 2.37 | 5.0 | 2.60 | - | 3.6 | 0.0 | 3.0 | - | 0.4 | 6.6 | 24 |
| #4 | 1072 | 26 | A1 | 2.35 | 5.1 | 2.58 | - | 2.9 | 0.0 | 3.4 | - | 0.8 | 6.3 | 14 |
| #5 | 1072 | 26 | A1 | 2.41 | 5.2 | 2.63 | - | 1.2 | 0.0 | 4.8 | - | 1.2 | 6.0 | 22 |
| #6 | 1072 | 25 | A2 | 2.37 | 4.5 | - | 2.08 | 5.6 | 1.0 | - | 1.0 | - | 6.6 | 25 |
| #7 | 1072 | 29 | A3 | 2.39 | 4.4 | - | 2.08 | 4.8 | 0.0 | - | 2.0 | - | 6.8 | 29 |
| #8 | 1072 | 29 | A4 | 2.43 | 4.5 | - | 2.11 | 3.5 | 0.0 | - | 3.0 | - | 6.5 | 29 |
| #9 | 1072 | 29 | A2 | 2.39 | 4.4 | 2.63 | 2.08 | 4.8 | 0.1 | 1.0 | 1.0 | 0.2 | 6.7 | 29 |
| #10 | 1072 | 29 | A2 | 2.39 | 4.5 | 2.62 | 2.10 | 3.9 | 0.1 | 1.6 | 1.0 | 0.3 | 6.5 | 17 |
| #11 | 1072 | 27 | A3 | 2.40 | 4.5 | 2.62 | 2.07 | 4.3 | 0.0 | 0.4 | 2.0 | 0.0 | 6.8 | 12 |
| #12 | 1072 | 29 | A1 | 2.35 | 4.6 | - | - | 7.2 | 0.5 | - | - | - | 7.2 | 29 |
| #13 | 1447 | 27 | B1 | 2.36 | 4.5 | 2.64 | - | 7.4 | 0.4 | 0.5 | - | 0.0 | 7.8 | 13 |
| #14 | 1447 | 27 | B1 | 2.34 | 4.5 | 2.65 | - | 5.9 | 0.1 | 2.0 | - | 0.3 | 7.9 | 27 |
| #15 | 1447 | 27 | B1 | 2.36 | 4.4 | 2.61 | - | 7.4 | 0.4 | 0.5 | - | 0.0 | 7.9 | 0 |
| #16 | 1447 | 24 | B2 | 2.34 | 4.5 | - | 2.06 | 6.7 | 0.2 | - | 1.0 | 0.1 | 7.7 | 24 |
| #17 | 1447 | 27 | B2 | 2.38 | 4.6 | - | 2.08 | 5.5 | 0.2 | - | 2.0 | 0.4 | 7.5 | 27 |
| #18 | 1447 | 25 | B2 | 2.39 | 4.3 | - | 2.15 | 5.6 | 0.1 | - | 2.6 | 1.0 | 8.1 | 14 |
| #19 | 1447 | 25 | B2 | 2.36 | 4.5 | 2.70 | 2.10 | 6.5 | 0.1 | 0.4 | 1.0 | 1.0 | 8.0 | 10 |
| #20 | 1447 | 25 | B2 | 2.36 | 4.4 | 2.71 | 2.11 | 5.9 | 0.1 | 0.8 | 1.0 | 0.6 | 7.6 | 3 |
| #21 | 1447 | 27 | B2 | 2.34 | 4.5 | 2.78 | 2.13 | 5.2 | 0.0 | 0.4 | 2.0 | 0.4 | 7.6 | 9 |
| #22 | 1447 | 27 | B1 | 2.36 | 4.4 | - | - | 7.8 | 0.8 | - | - | 0.0 | 7.8 | 27 |





**Table 3.** List of the simulation runs with their initial coordination complexes and complexes observed for at least 3 ps during the simulation. For runs where multiple complexes appear the observed complexes are listed in order of decreasing abundance.

| Run ID | simulation parameter | | Cell | complexes | |
| | $\rho$ (kg m$^{-3}$) | time (ps) | | initial | formed |
|---|---|---|---|---|---|
| #1 | 1072 | 25 | A1 | $[\mathrm{YCl(H_2O)_5}]^{2+}$ | $[\mathrm{YClOH(H_2O)_5}]^+$ , $[\mathrm{YCl(H_2O)_6}]^{2+}$ |
| #2 | 1072 | 23 | A1 | $[\mathrm{YCl_2(H_2O)_4}]^+$ | $[\mathrm{YCl_2(H_2O)_5}]^+ \cdot \mathrm{Na}^+$ , $[\mathrm{YCl_2(H_2O)_5}]^+$ , $[\mathrm{YCl_2OH(H_2O)_4}]_{aq}$ |
| #3 | 1072 | 24 | A1 | $[\mathrm{YCl_3(H_2O)_3}]_{aq}$ | $[\mathrm{YCl_3(H_2O)_4}]_{aq}$, $[\mathrm{YCl_3(H_2O)_4}]_{aq} \cdot \mathrm{Na}^+$ |
| #4 | 1072 | 26 | A1 | $[\mathrm{YCl_4(H_2O)_2}]^-$ | $[\mathrm{YCl_4(H_2O)_2}]^- \cdot \mathrm{Na}^+$ , $[\mathrm{YCl_4(H_2O)_2}]^-$ , $[\mathrm{YCl_3(H_2O)_3}]_{aq} \cdot \mathrm{Na}^+$ , $[\mathrm{YCl(H_2O)_6}]^{2+} \cdot \mathrm{Na}^+$ |
| #5 | 1072 | 26 | A1 | $[\mathrm{YCl_5(H_2O)}]^{2-}$ | $[\mathrm{YCl_5(H_2O)}]^{2-} \cdot \mathrm{Na}^+$ , $[\mathrm{YCl_5(H_2O)}]^{2-}$ , $[\mathrm{YCl_4(H_2O)_2}]^- \cdot \mathrm{Na}^+$ |
| #6 | 1072 | 25 | A2 | $[\mathrm{YF(OH)(H_2O)_5}]^+$ | $[\mathrm{YFOH(H_2O)_5}]^+$ |
| #7 | 1072 | 29 | A3 | $[\mathrm{YF_2(H_2O)_5}]^+$ , | $[\mathrm{YF_2(H_2O)_5}]^+$ |
| #8 | 1072 | 29 | A4 | $[\mathrm{YF_3(H_2O)_4}]_{aq}$ | $[\mathrm{YF_3(H_2O)_4}]_{aq}$ |
| #9 | 1072 | 29 | A2 | $[\mathrm{YClF(H_2O)_5}]^+$ | $[\mathrm{YClF(H_2O)_5}]^+$ , $[\mathrm{YClF(H_2O)_5}]^+ \cdot \mathrm{Na}^+$ |
| #10 | 1072 | 29 | A2 | $[\mathrm{YCl_2F(H_2O)_4}]_{aq}$ | $[\mathrm{YCl_2F(H_2O)_4}]_{aq} \cdot \mathrm{Na}^+$ , $[\mathrm{YClF(H_2O)_5}]^+$ , $[\mathrm{YClFOH(H_2O)_4}]_{aq} \cdot \mathrm{Na}^+$ |
| #11 | 1072 | 27 | A3 | $[\mathrm{YClF_2(H_2O)_4}]_{aq}$ | $[\mathrm{YF_2(H_2O)_5}]^+ \cdot \mathrm{Na}^+$ , $[\mathrm{YClF_2(H_2O)_4}]_{aq}$, $[\mathrm{YClF_2(H_2O)_4}]_{aq} \cdot \mathrm{Na}^+$ |
| #12 | 1072 | 29 | A1 | $[\mathrm{Y(H_2O)_7}]^{3+}$ | $[\mathrm{YOH(H_2O)_6}]^{2+}$ , $[\mathrm{Y(H_2O)_7}]^{3+}$ |
| #13 | 1447 | 27 | B1 | $[\mathrm{YCl(H_2O)_6}]^{2+}$ | $[\mathrm{YCl(H_2O)_7}]^{2+}$ , $[\mathrm{YOH(H_2O)_7}]^{2+}$ |
| #14 | 1447 | 27 | B1 | $[\mathrm{YCl_2(H_2O)_5}]^+$ | $[\mathrm{YCl_2(H_2O)_6}]^+$ , $[\mathrm{YCl_2(H_2O)_6}]^+ \cdot \mathrm{Na}^+$ |
| #15 | 1447 | 27 | B1 | $[\mathrm{YCl_3(H_2O)_4}]_{aq}$ | $[\mathrm{YCl(H_2O)_7}]^{2+}$ , $[\mathrm{YOH(H_2O)_7}]^{2+}$ , $[\mathrm{YCl_2(H_2O)_6}]^+$ |
| #16 | 1447 | 24 | B2 | $[\mathrm{YF(H_2O)_7}]^{2+}$ | $[\mathrm{YF(H_2O)_7}]^{2+}$ , $[\mathrm{YFOH(H_2O)_6}]^+$ |
| #17 | 1447 | 27 | B2 | $[\mathrm{YF_2(H_2O)_5}]^+$ | $[\mathrm{YF_2(H_2O)_6}]^+$ , $[\mathrm{YF_2OH(H_2O)_5}]_{aq}$ |
| #18 | 1447 | 25 | B2 | $[\mathrm{YF_3(H_2O)_4}]_{aq}$ | $[\mathrm{YF_3(H_2O)_5}]_{aq} \cdot \mathrm{Na}^+$ , $[\mathrm{YF_2(H_2O)_6}]^+ \cdot \mathrm{Na}^+$ , $[\mathrm{YF_2OH(H_2O)_5}]_{aq} \cdot \mathrm{Na}^+$ |
| #19 | 1447 | 25 | B2 | $[\mathrm{YClF(H_2O)_5}]^+$ | $[\mathrm{YF(H_2O)_7}]^{2+}$ , $[\mathrm{YClF(H_2O)_6}]^+ \cdot \mathrm{Na}^+$ , |
| #20 | 1447 | 25 | B2 | $[\mathrm{YCl_2F(H_2O)_5}]_{aq}$ | $[\mathrm{YFCl(H_2O)_6}]^+$ , $[\mathrm{YFCl(H_2O)_6}]^+ \cdot \mathrm{Na}^+$ , $[\mathrm{YFOH(H_2O)_6}]^+$ |
| #21 | 1447 | 27 | B2 | $[\mathrm{YClF_2(H_2O)_5}]_{aq}$ | $[\mathrm{YF_2(H_2O)_6}]^+ \cdot \mathrm{Na}^+$ , $[\mathrm{YClF_2(H_2O)_5}]_{aq} \cdot \mathrm{Na}^+$ , $[\mathrm{YF_2(H_2O)_5}]^+$ |
| #22 | 1447 | 27 | B1 | $[\mathrm{Y(H_2O)_8}]^{3+}$ | $[\mathrm{Y(H_2O)_8}]^{3+}$ , $[\mathrm{YOH(H_2O)_7}]^{2+}$ |





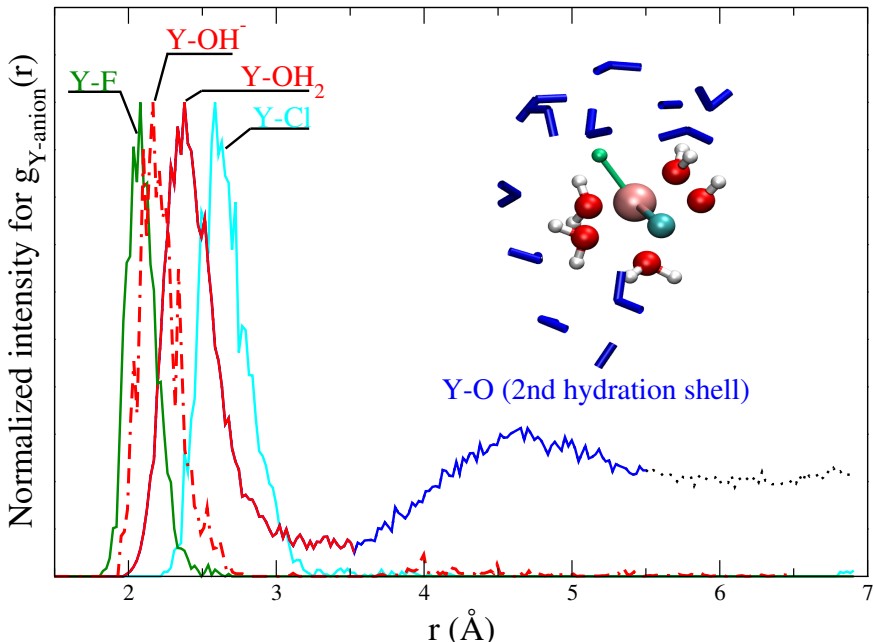

**Figure 3.** Radial distribution functions of Y-(Cl,O,F) scaled to the maximum of the $g_{ij}(r)$ from runs #9 and #12 together with a snapshot of a $[YClF(OH)(H_2O)_4]$ complex. In the snapshot (from a run at $1.3\,\text{GPa}$ and $800°\text{C}$) the central yttrium atom is surrounded by chlorine (cyan), fluorine (green), a hydroxyl group and water molecules (red–O and white–H balls). Water molecules of the $2^{nd}$ shell are shown as blue sticks. This visualization illustrates the relation between $g_{ij}(r)$ and the atomic structure of the aqueous species. The colors of the ligands in the snapshot are equivalent to those in the $g_{ij}(r)$ functions.

It corresponds to the $2^{nd}$ hydration shell, which is formed around all complexes at all studied $P/T$ conditions. For all fluid compositions, association of $NaCl_n$ (n=1-3) and of $Na^+$ with the yttrium complex is observed.

At $1.3\,\text{GPa}$, the average Y coordination by O, Cl and/or F is about seven (see Fig. 4) with two exceptions, runs #4 and #5, even if the initial coordination is lower (runs #1-3). In runs #1-3, the initial Y-Cl coordination is retained over the whole AIMD run time, whereas for #4 and #5 the fourfold and fivefold Cl-coordination do not persist and the time average yttrium

coordination is below seven. In #4 after the fourfold coordinated chloride complex is dissociated a total coordination number of seven is reached at the end of the simulation run. Frequently, the formation of $OH^-$ in the first hydration shell of yttrium is observed. The major hydroxide formation mechanism will be discussed below.

Figure 5 provides an overview over the formation and dissolution of selected structural units in the course of the simulations, i.e. the stability of the initial complexes, Y hydroxide association and bonding of the coordinating halogen to sodium. All five

chloride complexes associate with sodium. The strongest association is observed in #2-5 where the sodium is connected to one or two chloride ligands of the Y complex. This association increases with the number of halide ligands in these complexes. Furthermore, in #4 and #5 this association initiates the dissociation of $[YCl_4(H_2O)_2]^-$ and $[YCl_5(H_2O)]^{2-}$. Even larger

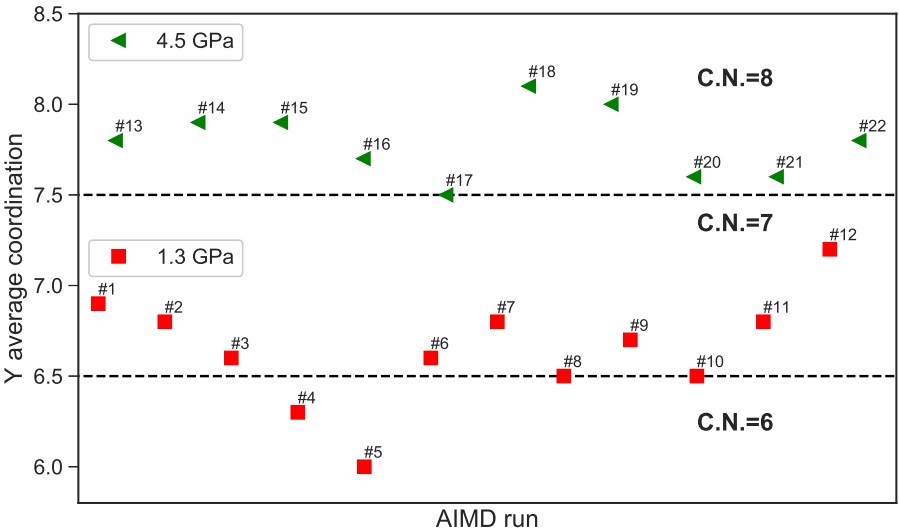

**Figure 4.** Average yttrium coordination by chloride, fluoride and oxygen for runs #1-22

clusters of sodium, constraint chlorides and the Y chloride complex appear over time scales of less than 3 ps. Moreover, from #1 to #5 the average Y coordination decreases with the increasing number of initial chloride ligands (Fig. 4). The Y-O distances

do not change significantly with the increasing number of chloride ligands from #1 to #4. Only in run #5 a significantly longer Y-O distance of 2.41 Å is observed. In #1, the highest amount of hydroxide is formed and $[\text{YClOH}(\text{H}_2\text{O})_5]^+$ is the major species. The Y chloride distances range from 2.58 Å in #1 to 2.63 Å in #5.

For pure Y fluoride complexes at the same conditions (#6-8) a slight increase of the Y-O distance with increasing number of fluoride ligands from 2.37 Å in $[\text{YF}(\text{H}_2\text{O})_6]^{2+}$ to 2.43 Å in $[\text{YF}_3(\text{H}_2\text{O})_4]_{\text{aq}}$ is observed. In all three runs the initial complex

persists over the whole simulation time. As in case of Y chloride solutions, the association of fluoride with sodium is observed but it is less pronounced. In #6, where only one fluoride is initially bonded to the central ion, $\text{OH}^-$ is formed within the first hydration shell of yttrium. The Y-F distances within the complexes are approximately 0.5 Å shorter than those of the Y-Cl species.

For the Y-(Cl,F) mixed complexes (runs #9-11) only the run starting initially from $[\text{YClF}(\text{H}_2\text{O})_5]^+$ does not show the

formation of multiple complexes over time. Here, only short separations of the $\text{Cl}^-$ over $\sim$1 ps from the complex occur. In #10 $[\text{YCl}_2\text{F}(\text{H}_2\text{O})_4]_{\text{aq}}$ dissociates to $[\text{YClF}(\text{H}_2\text{O})_6]^+$ after 11.5 ps. This complex is present over approximate 10 ps in conjunction with the formation of $[\text{YClFOH}(\text{H}_2\text{O})_4]_{\text{aq}}$ followed by the re-association of the initial complex. In #11 starting from $[\text{YClF}_2(\text{H}_2\text{O})_4]_{\text{aq}}$ the initially bonded chloride is released after $\sim$12 ps and $[\text{YF}_2(\text{H}_2\text{O})_5]^+$ is formed. The Y-(F,Cl) distances of the mixed complexes are comparable to those of the pure ones.



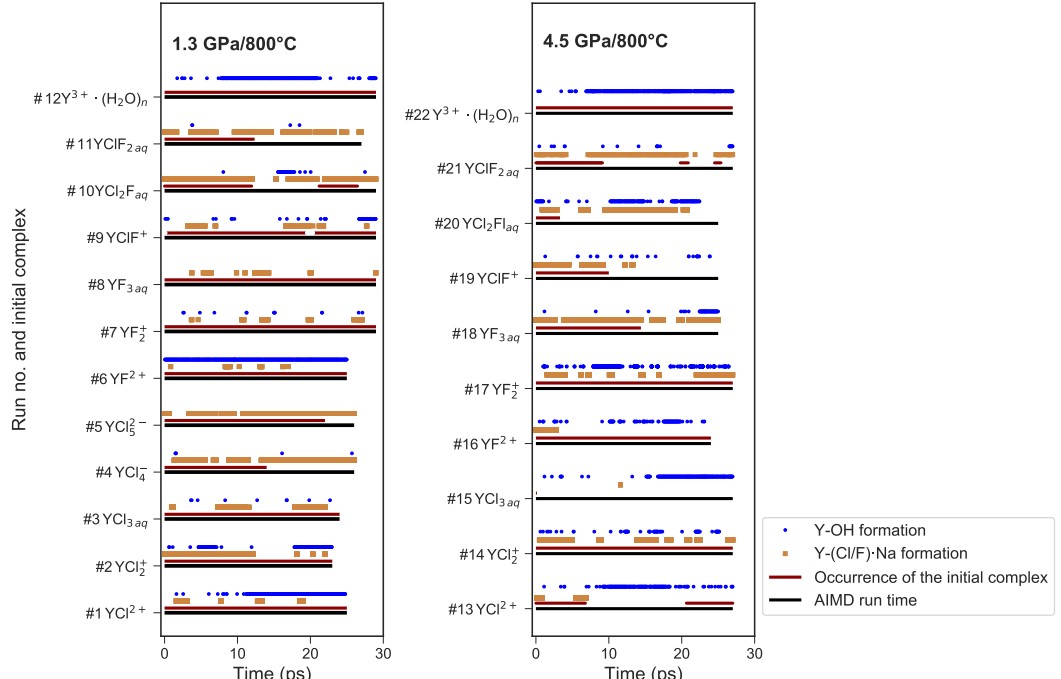

**Figure 5.** Presence of selected ion pairs or complexes during the different AIMD simulations. This includes the formation of OH$^-$ within the first hydration shell of yttrium, the association of sodium with the coordinating halogens and the stability of the initial complexes over AIMD time.

In run #12 starting from $[\mathrm{Y(H_2O)_7}]^{3+}$ hydroxide is formed within the first 8 ps (see Fig. 5), which results in the formation of $[\mathrm{YOH(H_2O)_6}]^{2+}$ that is present over 14 ps of the AIMD time followed by a reassociation and a redissociation, which suggests a dynamic change between these two species.

    In the high pressure runs at 4.5 GPa (#13-22) the average Y coordination is about eight (see Fig. 4). In case of the Y chloride complexes, the dissociation of the onefold and threefold coordinated complexes is observed in runs #13 and #15

(see Fig. 5). Only in run #14 the initial Y chloride complex $[\mathrm{YCl_2(H_2O)_5}]^+$ persists over the whole 27 ps trajectory. The higher coordinated Y chloride complexes break apart within the equilibration run and the results are not further analyzed. This breakdown is partly driven by the association of the coordinating chloride with sodium. For instance, in run #13 one sodium chloride unit associates with the Y complex before the chloride dissociates from the Y complex and $[\mathrm{NaCl_3}]^{2-}$ is formed for $\sim$3 ps (Fig. 6). The resulting $[\mathrm{Y(H_2O)_8}]^{3+}$ associates with OH$^-$ shortly afterwards. In all high $P$ runs, the formation of OH$^-$

by self-dissociation of H$_2$O close to the yttrium ion can be seen as in the low $P$ runs.

    Figure 6 illustrates the OH$^-$ formation mechanism as it evolves for Y chloride and Y fluoride complexes at low and high pressure conditions for the example of $[\mathrm{YCl(H_2O)_7}]^{2+}$ in run #13. After the initial complex dissociates within the first 7 ps into $[\mathrm{Y(H_2O)_8}]^{3+}$ a proton is transferred between one H$_2$O in the first hydration shell and a water molecule of the second shell





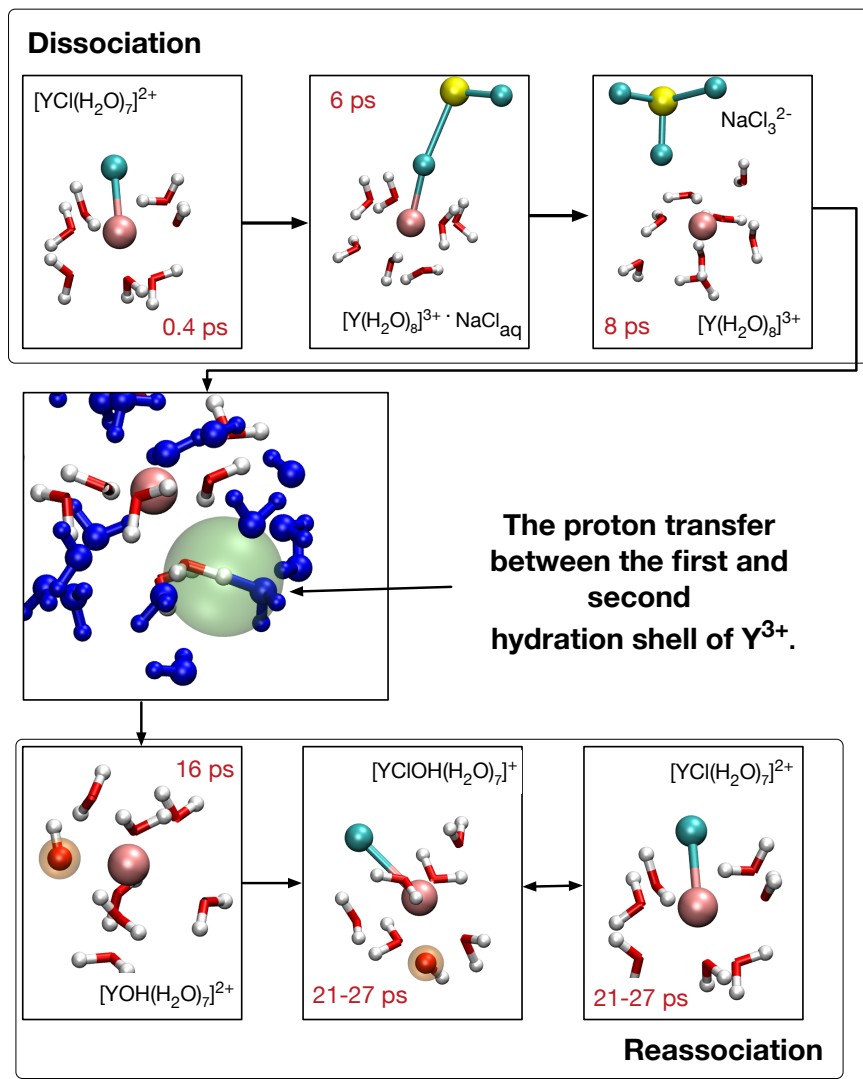

**Figure 6.** Formation of $\left[\mathrm{YOH(H_2O)_7}\right]^{2+}$ and re-association of initial complex $\left[\mathrm{YCl(H_2O)_7}\right]^{2+}$ in run #13. The blue colored water molecules are located in the $2^{nd}$ hydration shell, red-white $\mathrm{H_2O}$ and $\mathrm{OH^-}$ are bonded in the first coordination shell and sodium is colored yellow. Chloride ions are in cyan and the $\mathrm{Y^{3+}}$ ion is copper colored. The red numbers indicate the time progress in the AIMD simulation. In center the proton transfer state is highlighted with a green sphere.

after additional 2-3 ps. The resulting $\left[\mathrm{YOH(H_2O)_7}\right]^{2+}$ complex is present over 14 ps followed by reassociation with chloride.

The thus formed $\left[\mathrm{YClOH(H_2O)_6}\right]^+$ persists during the remaining simulation time with some interruptions due to short-lived proton transfers.





For the Y fluoride complexes at the high pressure conditions the onefold or twofold Y by F coordination persists over the whole simulation runs (#16, #17). All complexes show association with one or two sodium ions over several picoseconds but this interaction does no lead to a dissociation of fluoride from yttrium. In #18, the initially threefold coordinated complex

dissociates after 14.5 ps and $[\mathrm{YF_2(H_2O)_6}]^+$ is formed. None of the mixed Y-(Cl,F) complexes at 4.5 GPa is stable over the entire simulation run. In each of those runs all chloride ions are dissociated from the yttrium after at most 10 ps and pure Y-F or Y-(F,OH) complexes remain. As in the low pressure run for the pure hydrated $\mathrm{Y}^{3+}$ a hydroxide ion is observed in the first hydration shell for a significant amount of simulation time.

The nearest Y-(O,Cl,F) distances show only small variations between both pressure conditions, typically in the range of

0.01-0.03 Å for the stable complexes (Tab. 2). A closer look at the distances between oxygen of the second hydration shell and the yttrium ion (see Y-O($2^{\mathrm{nd}}$) in Tab. 2) reveals a continuous increase from the purely hydrated ion with increasing Cl coordination at low pressures from 4.7 to 5.2 Å. In all other cases these distances are rather similar in a range between 4.3 to 4.6 Å.

Comparing the average halide ion coordination by $\mathrm{H_2O}$ molecules, differences between purely hydrated halide ions or halide

ions associated to the yttrium ion as well as pressure-induced changes are observed (see Supporting Information Tab. S1). For the chloride ion at 1.3 GPa the number of hydrating water molecules increases from two to four between $\mathrm{YCl_2^+}$ - $\mathrm{YCl_{3\,aq}}$ and dissociated $\mathrm{Cl}^-$, whereas in the mixed complexes $\mathrm{YClF_{2\,aq}}$ the chloride hydration number is close to three. For $\mathrm{YCl^{2+}}$, $\mathrm{YCl_4^-}$ and $\mathrm{YCl_5^{2-}}$ this number lies between two and three. The fluoride ion is coordinated by one water molecule in the Y fluoride and mixed complexes. At 4.5 GPa, $\mathrm{F}^-$ is hydrated by four $\mathrm{H_2O}$ and by two when associated with the metal ion, whereas the

dissociated $\mathrm{Cl}^-$ coordination increases to five solvent molecules. For those Y chloride complexes that persist for at least 10 ps in the AIMD run, the chloride ion is coordinated by three water molecules.

To conclude this section, the main findings from the AIMD simulations are summarized. Firstly, the pure Y chloride complexes $\mathrm{YCl^{2+}}$, $\mathrm{YCl_2^+}$ and $\mathrm{YCl_{3\,aq}}$ do not dissociate within the course of the simulation of at least 23 ps at a pressure of 1.3 GPa ($\rho$ =1072 kg m$^{-3}$) but they do at higher pressure (4.5 GPa, 1447 kg m$^{-3}$) except for $\mathrm{YCl_2^+}$. Furthermore, $\mathrm{YCl_4^-}$ and $\mathrm{YCl_5^{2-}}$

are not stable in the unbiased AIMD simulations. Secondly, the pure Y fluoride complexes are stable at 1.3 GPa. It is important to state that at these conditions the formation of hydrofluoric acid is very strong when the fluoride is not associated to the metal ion. At 4.5 GPa the neutral complex $[\mathrm{YF_3(H_2O)_5}]_{\mathrm{aq}}$ dissociates and a lower coordinated species forms. Thirdly, $\mathrm{OH}^-$ is formed in the first hydration shell of Y chloride and Y fluoride complexes due to the self-dissociation of water. Its abundance increases with decreasing number halide ligands. Furthermore, chloride as well fluoride form mixed complexes with yttrium

and hydroxide at both $P/T$ conditions whereas mixed Y-(Cl,F) complexes are rather unstable. The overall coordination of yttrium changes from ~7 at 1.3 GPa to ~8 at 4.5 GPa.

## 3.2 Free energy exploration

Although several complexes are observed in some of the runs described above it is not feasible to derive the corresponding formation constants of the aqueous species directly from the AIMD simulations. This would require much longer simulation

times to ensure the statistical significance of the relative species abundance. On the tens of picoseconds timescale, we did not





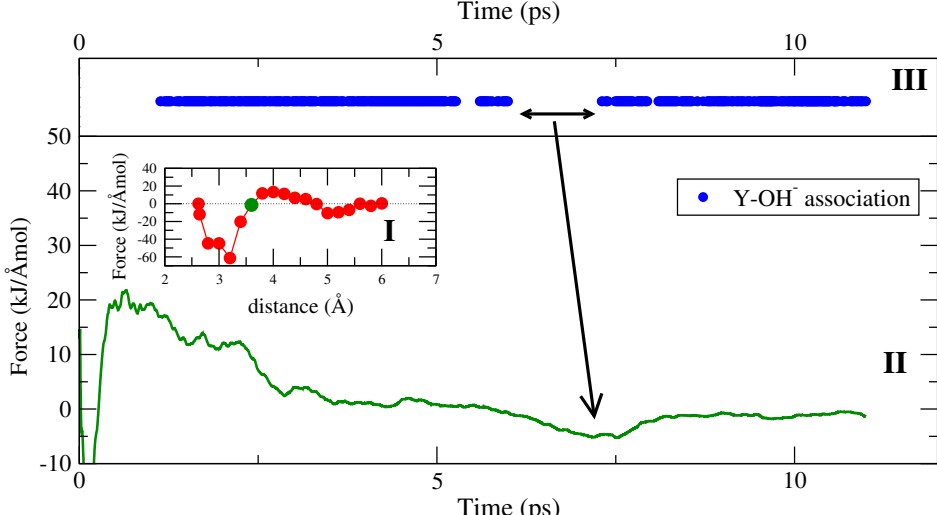

**Figure 7.** (I) Potential of mean force of TI-4. The green circle indicates the constraint distance of 3.6 Å for which the cumulative mean force is shown in (II). Dots in (III) indicate the presence of Y-OH$^-$ bonds during the simulation run.

observe a complete exchange of the halogen ligands of the Y chloride or Y fluoride complexes except for run #15. To overcome this problem we apply the thermodynamic integration (TI) approach within a constraint AIMD simulation to compute the reaction free energies of aqueous complex dissociation. As described in the previous section the mixed Y-(Cl,F) complexes have a tendency to dissociate already during the conventional AIMD runs, which indicates their low stability. Therefore, no TI

runs are performed for those complexes. When the constraint halide ion associates with hydrogen during the constraint MD the simulation is stopped and the integration step is repeated from a different starting configuration. Thus, we confirm that all results are reproducible within a series of simulations. The derived Helmholtz free energies ($\Delta_r A$) of the reactions R2 and R3 are listed in Tab. 4.

At 1.3 GPa, in TI-1 starting from $[YCl(H_2O)_6]^{2+}$ in nearly all TI steps close to the yttrium OH$^-$ formed by the hydrolysis

of water molecules within the first 2-4 ps. Therefore, the obtained dissociation energy of 36.1 kJ mol$^{-1}$ does not distinguish between $[YCl(H_2O)_6]^{2+}$ and $[YOHCl(H_2O)_5]^+$ complexes. For TI-2, we observe a similar dissociation energy of 38.8 kJ mol$^{-1}$ but much less hydroxide ions are formed such that in average over all integration steps Y-OH$^-$ appears in only 14 % of the total simulation time. The lowest dissociation energy of the Y chloride complexes at 1.3 GPa occurs in TI-3 with 26.4 kJ mol$^{-1}$. In TI-3 only little amounts of OH$^-$ are formed. While the integration proceeds the yttrium-oxygen coordination changes (in-

cluding OH$^-$ and H$_2$O) for all complexes at a constraint Y-Cl distance between 3.6-4.0 Å. The removed chloride as well the remaining Y chloride complex associate with sodium during the dissociation process for a few picoseconds. Dissociation energies of the Y fluoride complexes at 1.3 GPa could not be obtained due to the strong association of H$^+$ and F$^-$. The formation of hydrofluoric acid prevents the required long constraint Y fluoride bond distances for the integration of the PMF.



**Table 4.** List of the formation Gibbs free energies (kJ mol$^{-1}$) derived from *ab initio* thermodynamic integration. The error of the free energies is estimated with 5 kJ mol$^{-1}$.

| Nr | Reaction | Cell | Temperature (°C) | Pressure (GPa) | Y-(Cl,F) (Å) | $\Delta_r G$ | $\Delta_r G°$ | log $K$ |
|---|---|---|---|---|---|---|---|---|
| TI-1 | $[YOH]^{2+} + Cl^- = [YClOH]^+$ | A1 | 800 | 1.3 | 2.57 | -36.1 | -57.4 | 2.79 |
| TI-2 | $[YClOH]^+ + Cl^- + H_3O^+ = [YCl_2]^+ + 2H_2O$ | A1 | 800 | 1.3 | 2.57 | -38.8 | -57.6 | 2.80 |
| TI-3 | $[YCl_2]^+ + Cl^- = [YCl_3]_{aq}$ | A1 | 800 | 1.3 | 2.64 | -26.4 | -41.8 | 2.03 |
| TI-4 | $[YOH]^{2+} + Cl^- = [YClOH]^+$ | B1 | 800 | 4.5 | 2.62 | -29.6 | -57.4 | 2.79 |
| TI-5 | $[YClOH]^+ + Cl^- + H_3O^+ = [YCl_2]^+ + 2H_2O$ | B1 | 800 | 4.5 | 2.64 | -8.5 | -34.4 | 1.68 |
| TI-6 | $[YOH]^{2+} + F^- + H_3O^+ = [YFOH]^+ + HF_{aq} + H_2O$ | B5 | 800 | 4.5 | 2.12 | -45.9 | -85.6 | 4.17 |
| TI-7 | $[YF]^{2+} + F^- = [YF_2]^+$ | B5 | 800 | 4.5 | 2.13 | -38.5 | -74.7 | 3.64 |
| TI-8 | $[YF_2]^+ + F^- = [YF_3]_{aq}$ | B5 | 800 | 4.5 | 2.14 | -36.3 | -72.5 | 3.53 |





For the 4.5 GPa runs, the dissociation energies of the Y chloride complexes significantly decrease. As in the lower pres-
sure runs, the integration does not distinguish between complexes in which OH$^-$ is present or absent. In TI-4, starting from
$[YClOH(H_2O)_6]^+$ and forming $[YOH(H_2O)_7]^{2+}$ a free energy difference of $29.6\,\mathrm{kJ\,mol^{-1}}$ is obtained. As illustrated in Fig. 7
the reassociation of OH$^-$ with an excess proton leads to a change of the constraint force due to the decreasing attraction of the
metal cation to the constraint chloride ligand. The most frequent Y-OH$^-$ association is observed at constraint Y-Cl distances
above 3.0 Å.

TI-5 yields the lowest dissociation energy of $8.5\,\mathrm{kj\,mol^{-1}}$ where in several of the integration steps the second chloride ligand
also dissociates and $[YOH(H_2O)_7]^{2+}$ is formed. The Y-Cl dissociation is preceded by an Y-OH$^-$ association and results in
the formation of $[YOH_{0-1}(H_2O)_{6-7}]^{3-2+}$. In those cases, the initial complex was reset and the integration step was restarted.
For $[YCl_3(H_2O)_5]_{aq}$ it was not possible to derive a dissociation energy because the initial complex dissociated at short Y-Cl
constraint distances within the first picosecond of each simulation.

In TI-6 to TI-8 the dissociation energies of Y fluoride complexes following reaction R3 are investigated. The equilibrium
distance between the yttrium ion and its fluoride ligands is smaller than the Y-Cl bond distance. Due to this shorter distance the
integration range to reach the dissociated state was reduced to 5.0 Å. As shown in Fig. 8 (and Fig. S1) the convergence of the
free energy is still reached. In each run of this simulation series we observe the temporary formation of hydrofluoric acid by the
protolysis from $H_3O^+$ to one of the constraint F$^-$and, compared to the low pressure runs, to a lesser extent by the protolysis
of $H_2O$.

The dissociation energies (Fig. 8) are quite similar between the Y fluoride complexes. During the integration, the Y com-
plexes as well the removing fluoride ions interact with sodium. No self-dissociation of the complexes is observed except for
$[YF_3(H_2O)_5]_{aq}$. In the latter case at a constant distance of 2.6 Å one of unconstrained fluorides separates from the initial
complex. However, this behavior is not reproducible.

During all integration runs at both pressures, the Y-O coordination number increases in average by one during the transfor-
mation from the associated to the dissociated state. For the Y chloride complexes the increase in hydration happens at a Y-Cl
distance between 3.6-4.1 Å, which is in the range of the repulsive part of the constraint force. For the dissociation of Y fluoride
complexes this distance decreases to 3.4-3.6 Å. A significant influence of the second hydration shell on the constraint force
of the reaction coordinate is not observed. All complexes and the removing halide ions interact with sodium. This interaction
cannot be quantified by the PMF but the association of the yttrium complex with sodium decreases with the number of halide
ligands.

### 3.3 Thermodynamic data

Finally, the reaction free energies derived from thermodynamic integration are transformed into standard state properties by
applying the activity corrections described in the methods section above. The standard state correction yields $\Delta_r G^\circ$ that are
significantly smaller compared to $\Delta_r G$ (see Tab. 4). This treatment does not consider explicitly the formation of HCl and HF
as well the association of yttrium with hydroxide because their formation during TI is not systematic and it is not possible
to quantify their contributions to the reaction free energies. As these contributions seem not to be negligible as shown in
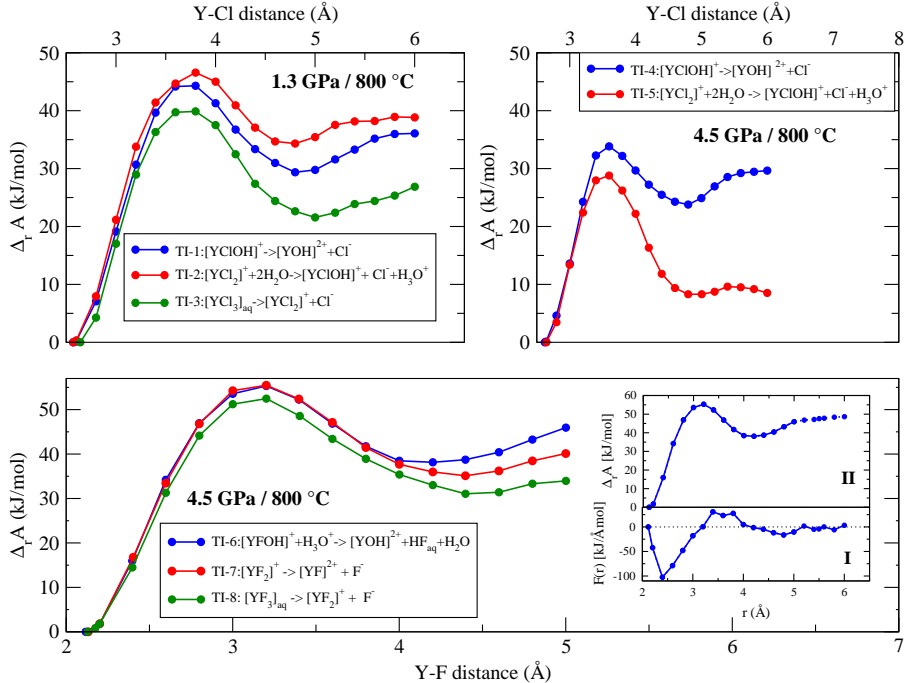

**Figure 8.** Evolution of the Helmholtz free energy derived from thermodynamic integration for Y-Cl/F complexes at $800^\circ$C and $1.3/4.5$ GPa Inset: potential of mean force of the dissociation reaction of $[\text{YFOH}]^+$ to $\text{YOH}_2^+$ for a integration length of $5.0$ Å and $6.0$ Å. (II) resulting Helmholtz free energy along the integration pathway (for higher magnification see Fig. S1).

Fig. 7 the logarithmic stability constants include not only the reactions listed in Tab. 5 and are therefore indicated with a star ($\log \beta^*$). Figure 9 shows the $\Delta_r G^\circ$ and $\log \beta^*$ for the different species. While pressure does not affect the formation reaction

of $\text{YCl}^{2+}$, the stability of $\text{YCl}_2^+$ decreases substantially with increasing pressure. Comparing the stability constants of chloride and fluoride species at $4.5$ GPa, the fluoride species are more stable by $1.4$ ($\log \beta_1^*$) to $3.3$ ($\log \beta_2^*$) log units.

# 4  Discussion

## 4.1  Molecular structure of the aqueous complexes

As mentioned in the introduction the number of studies focusing on the hydration environment of yttrium or other HREE
in aqueous fluids at high $T$ and $P$ conditions is very limited. The average coordination of seven nearest neighbors that is observed in the simulations at $1.3$ GPa fits in the range of experimental results. Vala Ragnarsdottir et al. (1998) observed 8-9 nearest neighbors at lower $T$ of $250^\circ$C and vapour pressures in highly concentrated chloride solution. But they did not find an association of Y chloride, whereas Mayanovic et al. (2002) reported a strong association with an average coordination of four at $500^\circ$C. The present simulations predict that $\text{YCl}_4^-$ is not stable at high $P/T$ conditions. The reason for this could be





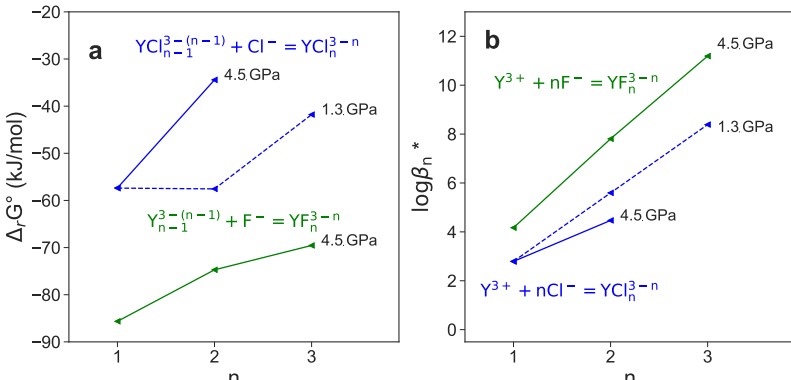

**Figure 9.** (a) Reaction Gibbs free energy $\Delta_r G^\circ$ of the different formation reactions, (b) change of the logarithmic stability constant of the different Y-(F,Cl) complexes.

**Table 5.** List of the logarithmic stability constants $\log \beta^*$ derived from *ab initio* thermodynamic integration in comparison to theoretical predictions based on HKF regression.

| density (kg m$^{-3}$) | $P$ (GPa)$^\top$ | reaction | $\log \beta^*$ | $\log \beta^\dagger$ |
|---|---|---|---|---|
| 1072 | 1.3 | $Y^{3+} + Cl^- = YCl^{2+}$ | 2.8 | 3.4[M] |
| 1072 | 1.3 | $Y^{3+} + 2Cl^- = YCl_2^+$ | 5.6 | 4.6[M] |
| 1072 | 1.3 | $Y^{3+} + 3Cl^- = YCl_{3\,aq}$ | 8.4 | - |
| 1447 | 4.5 | $Y^{3+} + Cl^- = YCl^{2+}$ | 2.8 | -1.7[M] |
| 1447 | 4.5 | $Y^{3+} + 2Cl^- = YCl_2^+$ | 4.5 | 0.04[M] |
| 1447 | 4.5 | $Y^{3+} + F^- = YF^{2+}$ | 4.2 | 3.4[L] |
| 1447 | 4.5 | $Y^{3+} + 2F^- = YF_2^+$ | 7.8 | 10.9[L] |
| 1447 | 4.5 | $Y^{3+} + 3F^- = YF_{3\,aq}$ | 11.2 | - |

* indicates that listed values involve further transition states that cannot be separated during the TI, $^\top$
pressure estimates assuming a 2 molal NaCl solution using the empirical equation of state by
(Mantegazzi et al., 2013), $^\dagger$ values computed using the DEW model (Sverjensky et al., 2014) with HKF
parameters reported by Loges et al. (2013)[L] for Y and Migdisov et al. (2009)[M] for Ho complexes

the too low average Y coordination in $YCl_4^-$ and $YCl_5^{2-}$. An average coordination of seven as present in the stable Y chloride complexes of the simulations cannot be achieved in these highly chlorinated species due to steric constraints. The Y-O distances derived from EXAFS spectra by Vala Ragnarsdottir et al. (1998) in the range of 2.36-2.39 Å are in good agreement with the atomic distances from the presented simulations while the conference abstract by Mayanovic et al. (2002) does not comprise quantitative data. Experiments and simulations are only partly comparable as the simulations were performed at higher $T$

(800 °C) than the experiments by Vala Ragnarsdottir et al. (1998) or Mayanovic et al. (2002).



The formation of stable Y-Cl species at 1.3 GPa and 4.5 GPa is consistent with observations by e.g. Tropper et al. (2011); Schmidt et al. (2007b) that the mobility of yttrium increases with increasing availability of $Cl^-$ in aqueous systems. The Y-Cl complexes become less stable with increasing $P$. The destabilization of metal-halide species with rising pressure is known from a variety of systems (see overview by Manning (2018)). This is commonly explained by the increase of the dielectric

constant with increasing density at constant $T$. The increase of $\epsilon$ ($\epsilon$=17.1 at 1072 kg m$^{-3}$ and $\epsilon$=26.2 at 1447 kg m$^{-3}$ according to Sverjensky et al. (2014)) leads to a stronger hydration of the metal ion by $H_2O$ and the stabilization of charged species. Therefore, $YCl_{3\,aq}$ is not stable at 4.5 GPa. The direct competition of both halide ligands in the mixed complexes shows a clear preference of yttrium to form stable bonds with $F^-$ rather than with $Cl^-$. Moreover, the lower reaction Gibbs free energies of the Y fluoride species in Fig. 9 (a) strongly support this observation.

Figure 10 shows a comparison of the Y-OH$^-$ formation between both pressure conditions in those aqueous complexes that are stable over at least 10 ps in the unconstrained AIMD simulations. A higher abundance of hydroxide groups is observed for Y-(Cl/F) complexes at lower $P$. Furthermore, the amount of formed OH$^-$ decreases with increasing number of ligands and is particularly high for the purely hydrated $Y^{3+}$ at both pressures. This observation cannot be explained by the increased self-dissociation of water, which increases with pressure (see e.g. Rozsa et al. (2018); Goncharov et al. (2005)). According to

Marshall and Franck (1981) the OH$^-$ molality is in the range of $10^{-6}$ to $10^{-5}$ under the investigated $P/T$ conditions. Therefore, the hydroxide formation in the simulation must be driven by charge compensation in the absence of other ligands around the yttrium. The low abundance of Y-OH$^-$ in the neutral species (e.g. $YF_{3\,aq}$) supports this interpretation. It has to be stressed that the initial simulation cells do not contain excess hydroxide ions. Therefore, the Y-OH$^-$ association is expected to be much higher if the OH$^-$ concentration increases. But it has to be underlined that the values in Fig. 10 are rather imprecise because

an equilibrium distribution of Y hydroxide bonds cannot be achieved in the rather short simulation time.

The association of yttrium with hydroxide as observed in the simulations was also noticed in high $P/T$ solubility experiments by Tropper et al. (2011) in NaCl brines but not in the $H_2O-NaF$ system (Tropper et al., 2013). The authors explained the association by a geometrical effect. The smaller HREE (in comparison to a LREE) have less attraction to a large chloride ion due to the so called "steric hindrance" as discussed by Mayanovic et al. (2009). But in case of the Y fluoride complexes

especially for $YF^{2+}$ the OH$^-$ formation is also controlled by the protolysis of $H_2O$ close to the metal ion. Therefore, the geometrical explanation does not hold to explain the Y hydroxide bonding. That this process was not detected in the experiments by Tropper et al. (2013) might be caused by the high fluoride content in the experiments. The majority of $YF_2^+$ in the experiments underline this conclusion. That the formation of Y-(F/OH) species was not detected in solubility experiments by Loges et al. (2013) up to 250 °C indicates that the protolysis of vicinal water by yttrium is a high temperature process. Protolysis at

high temperatures was also reported by van Sijl et al. (2010) for $Ti^{4+}$ and might be a general property of high-field-strength elements.

Entropy is an additional driving force for the ion association due to changes in the local solvent structure near the critical point and above as discussed, e.g., by Sherman (2010) and Mei et al. (2014) based on AIMD simulations. The dominant effect arises from the translational entropy through hydration changes of the ions during the aqueous reaction. Such a concept was

already discussed by Mesmer et al. (1988). They proposed that the change in the electrostriction volume of the solvent controls





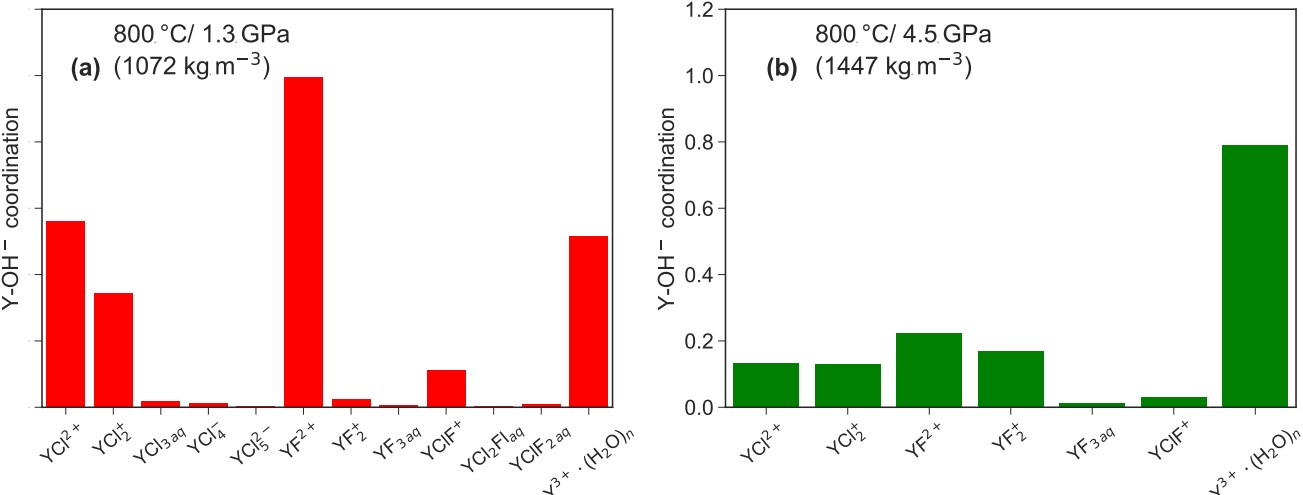

**Figure 10.** Comparison of the average Y-OH$^-$ coordination number between the different complexes that exist for at least over 10 ps in the unconstrained AIMD at 1.3 GPa (a) and 4.5 GPa (b).

the association of aqueous species, e.g. $NH_{3\,aq}, HCl_{aq}$ and $NaCl_{aq}$. In the present simulation study a relation between the change of hydration of chloride and the stability of certain complexes is observed at 1.3 GPa. For example the formation of $YCl_2^+$ and $YCl_{3\,aq}$ (TI-2, TI-3) releases two inner-sphere solvation water molecules because the hydration of $Cl^-$ decreases from four to two. This may explain the very similar reaction Gibbs energy of -38.8 kJ mol$^{-1}$ in TI-2 compared to TI-1 (-

36.1 kJ mol$^{-1}$) where only one $H_2O$ is released (besides the effect of $OH^-$ formation). Normally, one would expect a decrease in released reaction free energy with increasing number of ligands. Due to the spontaneous formation of HF the number of $H_2O$ in the hydration shell of an $F^-$ ion cannot be derived directly from the 1.3 GPa simulations. At 4.5 GPa two hydration $H_2O$ are exchanged for $F^-$ and $Cl^-$ during the dissociation of the initial complexes. This convergence of the hydration change for the halogens supports the assumptions by Mesmer et al. (1988) and Mei et al. (2015) that the entropic effect decreases with

increasing density at constant temperature ($\geq 200\,°C$).

## 4.2 Comparison of the thermodynamic data with HKF regression

Experimental high $P$ and high $T$ thermodynamic properties of Y chloride and fluoride species are not available to compare and evaluate the present simulation results. Therefore, the Deep Earth Water (DEW) model (Sverjensky et al., 2014) is used to compute stability constants of $YF^{2+}$ and $YF_2^+$ derived from solubility experiments up to $250\,°C$ by Loges et al. (2013)

using a HKF regression to $800\,°C$ and a fluid density equal to that of the simulations. There are no stability data of Y chloride complexes available but due to the similarities of yttrium and holmium (Ho) chloride complexes at room temperature (Luo and Byrne, 2001) the behavior of Y chloride complexes is assumed to be similar to Ho chloride complexes (Migdisov et al., 2019).





The Ho chloride HKF parameters are taken from Haas et al. (1995); Migdisov et al. (2009). In addition, Y/Ho-OH$^-$ stabilities are derived from data of Shock et al. (1997); Haas et al. (1995). The results are shown in Fig. 11.

Comparing the results in Fig. 11 (a) it can be observed that the stabilities of YCl$^{2+}$ and YCl$_2^+$ are similar to those of the Ho chloride species within approximately one logarithmic unit. For $\beta_3$ (Y−Cl), Y and Ho show opposite behavior. In this case an increase in the stability of YCl$_{3\,aq}$ is observed whereas the Ho $\beta_3$ (Ho-Cl) decreases (Haas et al., 1995). Migdisov et al. (2009) do not report any $\beta_3$ (Cl) values. For Y and Ho hydroxide species the stabilities are in the range of YCl$^{2+}$. At higher density (Fig. 11 (b)) the HKF model does not yield stable Ho/Y chloride species whereas the AIMD does for YCl$^{2+}$ and YCl$_2^+$. The

formation of Y-OH$^-$ in the AIMD runs suggests that Y-OH$^-$ association may occur in high density brines.

Due to the lack of AIMD $\log \beta_n$ (Y−F) data, Fig. 11 (c) only shows values derived from the HKF parameters. Here, the Y fluoride species (Loges et al., 2013) have the highest stability compared to Ho fluoride (Migdisov et al., 2009; Haas et al., 1995), Y hydroxide (Shock et al., 1997) and Ho hydroxide (Haas et al., 1995) species. It should be mentioned here that the model from (Haas et al., 1995) is suspected to overestimate the HREE-F stability (Migdisov and Williams-Jones, 2014). In

Fig. 11 (d) it is shown that for the higher fluid density the $\beta_1^*$ (Y-F) values from the simulations are consistent with regression based on experimental data (Loges et al., 2013) within one log unit. The formation constant of YF$_2^-$ from Loges et al. (2013) is higher compared to the AIMD results. On the other hand, the Ho-fluoride (Migdisov et al., 2009) complexes and the Y/Ho-hydroxide complexes have a much lower stability. Those differences indicate a different behavior of the geochemical twins Y and Ho in fluoride-rich environments, which could explain the fractionation of these elements even at high $P/T$ conditions as

it was observed e.g. in hydrothermal systems (Bau and Dulski, 1995) and discussed by Loges et al. (2013). Furthermore, the comparable stability of both Y chloride and Ho chloride complexes confirms the comparable geochemical behavior of the ions in chloride-rich solutions as assumed by Migdisov et al. (2019).

In general, at lower fluid densities we note similar stabilities of Y and Ho chloride complexes. The strong divergence for the neutral complexes might be explained by the origin of the HKF parameters. The data from Haas et al. (1995) were derived

from thermodynamic predictions based on measurements at $25\,^\circ$C and $1\,$bar where only a very small amount of a neutral species is formed and the uncertainty of the extrapolation is large. This interpretation is supported by reported experiments of Migdisov et al. (2009) and Loges et al. (2013) who do not observe the formation of neutral complexes up to $250\,^\circ$C. Only *in situ* measurements by (Mayanovic et al., 2002) show higher Cl coordination of yttrium whereas for Ho no data is available.

## 4.3   The role of chloride and fluoride for the mobilization of Y$^{3+}$ at subduction zone conditions

Stable Y chloride and Y fluoride complexes are found over the investigated $P/T$ range. The fluoride-bearing species are more stable than the chloride species. This outcome is consistent with a variety of studies from the field (see e.g. Newton et al. (1998)) as well as from experiments (Hetherington et al., 2010; Tsay et al., 2014; Tropper et al., 2013). Figure12(a-c) shows the distribution of the Y fluoride and chloride complexes as a function of the dissolved salt concentration. In Fig. 12 (d) the speciation change as function of the F$^-$concentration is shown. Beside the presented thermodynamic data from this study

competing aqueous reactions as the formation of HCl$_{aq}$, NaCl$_{aq}$, NaF$_{aq}$, HF$_{aq}$, NaOH$_{aq}$ as expected in high grade metamorphic fluids (Aranovich and Safonov, 2018; Galvez et al., 2016; Manning, 2018)) and observed in the simulations are included.





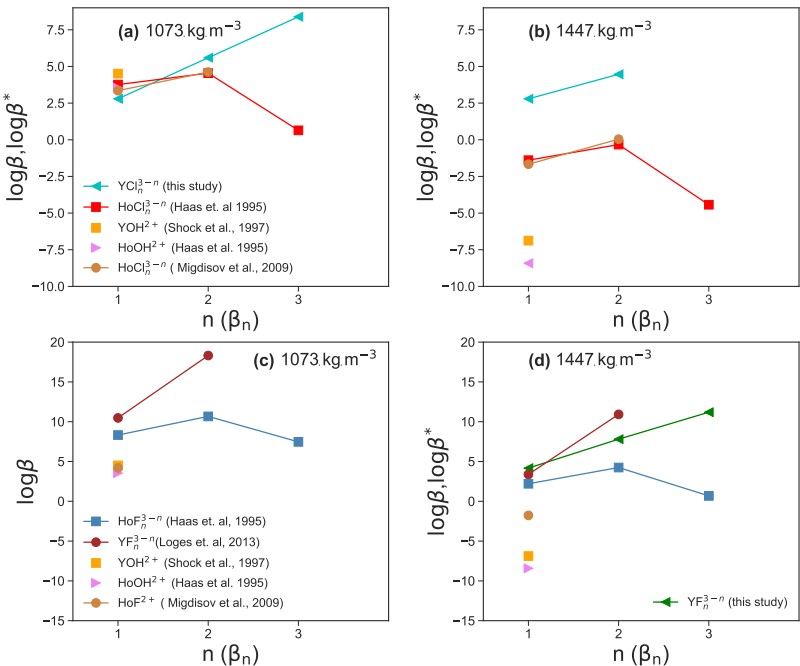

**Figure 11.** Comparison of the aqueous species stability derived from the AIMD simulation and HKF regression using the DEW model (Sverjensky et al., 2014) at $800°C$ and a fluid density of $1072\,\mathrm{kg\,m^{-3}}$ and $1447\,\mathrm{kg\,m^{-3}}$.

Possible complexation with silica components due to the lack of thermodynamic data as well as mineral reactions are not included. The neutral $pH/pOH$ is fixed by the ion product of water (Marshall and Franck, 1981). The thermodynamic properties of the competing aqueous species at high $P/T$ conditions are computed using the HKF model parameters reported by Mei et al. (2018); Shock and Helgeson (1988); Shock et al. (1997) and Shock et al. (1989) using the DEW model (Sverjensky et al., 2014). For this simple model, a Y concentration of 23 ppm in solution is assumed. This amount corresponds to measurements in subducted material such as mid-ocean-ridge basalt (Sun and McDonough, 1989) and is in the range of Y solubility in $Cl^-$-bearing brines reported by Tropper et al. (2011); Schmidt et al. (2007a).

Comparing the Y chloride speciation in Fig. 12 (a,b) it can be noted that yttrium is mainly associated with $Cl^-$ at an NaCl concentration of $\sim 0.005\,\mathrm{molal}$ in the lower density fluid whereas at the higher density conditions this state is reached at $\sim 0.01\,\mathrm{molal}$ NaCl in solution. For a fluid density of $1072\,\mathrm{kg\,m^{-3}}$ $YCl_3$ is the main species above an NaCl concentration of $0.01\,\mathrm{molal}$. At high density conditions $YCl_2^+$ is the major species at a higher NaCl concentration of $0.14\,\mathrm{molal}$. This shows that rather small amounts of dissolved NaCl are needed to form stable Y chloride species at subduction zone $P/T$ conditions if the yttrium is in solution. The required amounts of chloride presumably occurs in subduction fluids as demonstrated in a


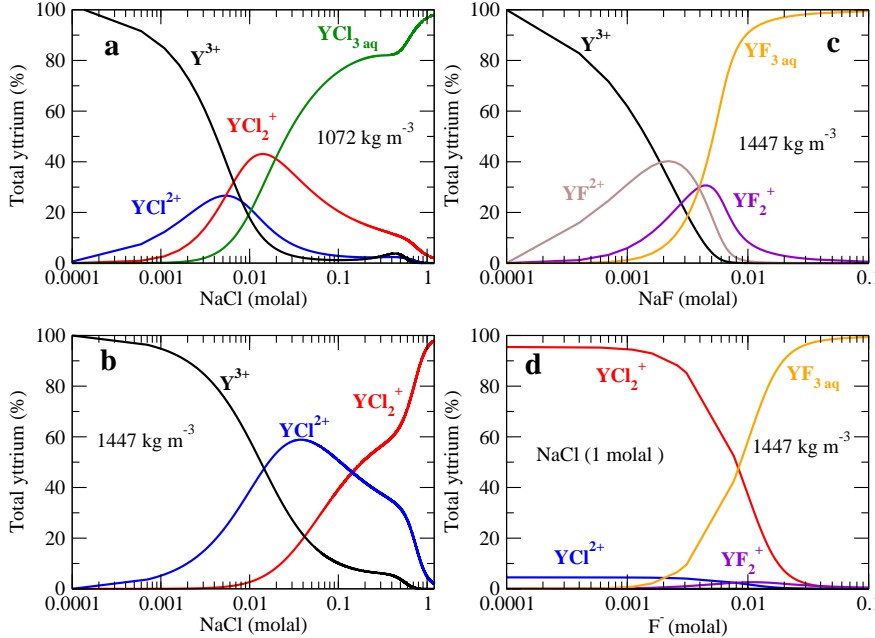

**Figure 12.** Y chloride and fluoride species distribution computed from the AIMD data assuming $23\,\mathrm{ppm}$ dissolved yttrium in aqueous solution at $800^\circ\mathrm{C}$. (a,b) Y chloride species distribution versus the logarithmic sodium chloride concentration in low and high density fluid, (c) Y fluoride species versus the amount of dissolved NaF and (d) Y-Cl/F species in a $1\,\mathrm{molal}$ NaCl solution with increasing $\mathrm{F}^-$ concentration. The $pH$ of the solutions is changing in a range of $\pm1.7$ over the concentration ranges.

variety of studies (Barnes et al., 2018; Aranovich and Safonov, 2018). However, it has been discussed by Tropper et al. (2013) that fluoride aqueous complexes are much more capable to mobilize significant amounts of yttrium. As shown in Fig. 12 (c) for a fluid bearing $\mathrm{Y} + \mathrm{NaF}$ (in the absence of other ligands) at an $\mathrm{NaF}$ concentration of about $0.0014\,\mathrm{molal}$, Y fluoride complexes are the majority species. Deep aqueous fluids in the Earth crust are presumed to be mixtures of $\mathrm{H_2O}$+salt (mainly NaCl, $\mathrm{CaCl_2}$, KCl) (Manning, 2018). Therefore, in Fig. 12 (d) the speciation in a $1\,\mathrm{molal}$ NaCl brine with increasing amount

of fluoride in the solution at a density of $1447\,\mathrm{kg\,m^{-3}}$ is computed. Here, $\mathrm{YCl_2^+}$ and $\mathrm{YF_{3aq}}$ are the dominant species. Below a $\mathrm{F}^-$ concentration of approximately $0.01\,\mathrm{molal}$, Y chloride complexes are formed and above this concentration Y fluoride becomes more important, at least in high density brines. At lower densities the formation of Y fluoride evolves at lower fluoride concentration due to the strong increase of complex stability as shown in Fig 11 (c).

     Estimates of the fluoride content in aqueous phases evolving in subducting slabs are given in the range of $\sim$0.003-0.18 molal

derived from analysis of melt inclusions and metamorphic rocks (Portnyagin et al., 2007; Hughes et al., 2018; Aranovich and Safonov, 2018), thermodynamic modeling (Zhu and Sverjensky, 1991) and $f\mathrm{HF}/f\mathrm{H_2O}$ ratio based on partitioning data between mineral and fluid (Sallet, 2000; Yardley, 1985). The simple model outlined above shows that only a low fluoride





concentration in the metamorphic aqueous fluids is needed to change the yttrium speciation and therefore its mobility. This outcome is in line with recent thermodynamic modeling by Xing et al. (2018) that shows that a small amount of dissolved
$F^-$ (tens of ppm) in ore forming solutions in equilibrium with a granite assemblage is able to mobilize significant amount of REE. But the high immobility of yttrium in crustal as well subduction zone metasomatism (Ague, 2017) might reflect a low fluorine activity in the majority of metasomatic fluids due to formation of fluoride species, e.g. $HF_{aq}$, $HF_2^-$, $SiF_6^{2-}$ or $Si(OH)_2F_{2\,aq}$. Therefore the included thermodynamic properties of HF from Shock et al. (1989) might underestimate its stability. A very low solubility of yttrium-bearing minerals as suggested by Migdisov et al. (2016) and the retrograde solubility
of REE phosphates (Schmidt et al., 2007a; Cetiner et al., 2005) could play an important role in the fixation of yttrium and other HREE.

At high densities, the stability of Y hydroxide complexes might be higher than the HKF regression indicates. As shown by Liu et al. (2012), at room temperature the Y-OH$^-$ complexation is sensitive to $pH$ changes and could evolve under more alkaline conditions as presumed for deep metasomatism (Galvez et al., 2016). But with the applied methods in this study no further
statement can be made due to the limitations of the PMF thermodynamic integration approach to extract reaction free energies for all relevant individual reactions including different intermediate states and dynamic changes such as proton transfer. To overcome this problem multiple constraints could be applied as suggested by Ivanov et al. (2006) but this would lead to even longer simulation times and might require more than the available computation resources for reaching sufficient convergence of structures and energies (Mark et al., 1994). Therefore, other free energies sampling methods could be promising alterna-
tives, e.g. the metadynamics approach (Laio and Parrinello, 2002). This method has been successfully applied to compute acid constant ($pK$s) in combination with quantum mechanical molecular dynamics (e.g. Sakti et al. (2018); Tummanapelli and Vasudevan (2015)) and to find new reaction pathways (see Pérez de Alba Ortíz et al. (2018); Pietrucci et al. (2018)). To probe aqueous systems or mineral-fluid interactions of more realistic size and composition classical molecular dynamics simulations in combination with force field interaction potentials or/and machine learning potentials (Behler and Parrinello, 2007; Cheng
et al., 2019) certainly have potential to provide significant progress in this field.

## 5   Summary and conclusion

The results of the *ab initio* molecular dynamics simulations provide new insight into the Y-Cl, Y-F and Y-OH$^-$ complexation in highly saline solutions as they occur in geological settings, e.g. of subduction zones. Firstly, Y chloride aqueous complexes are observed at $800\,°C$ and 1.3 or 4.5 GPa where $YCl_3$ and $YCl_2^+$ are the major species. Moreover, the destabilization of $YCl_4$
and $YCl_5$ indicates that there are no other Y chloride species that have to be considered at least in high grade metamorphic processes.

The extracted thermodynamic properties of Y chloride species presented in this study are the first published data set to our knowledge. The stability of Ho chloride complexes derived from thermodynamic calculations based on an HKF regression (Migdisov et al., 2009) at a solution density of $1073\,kg\,m^{-3}$ suggests that Y and Ho behave very similar in Cl$^-$ rich solutions
but with increasing solution density Y chloride complexes are more favorable than Ho chloride complexes. On the contrary,

Y shows a much stronger association with fluoride compared to Ho, which could explain their different behavior in F-rich aqueous environments. A different association behavior of both elements with respect to $OH^-$ would have an even higher impact on the Y/Ho fractionation because mineral solubilities and mineral surface adsorption are much more controlled by the $pH$ and $pOH$ values than the halide content of the aqueous fluid.

The formation of Y fluoride complexes in high density aqueous fluids happens even at very low $F^-$ concentrations and should lead to a high mobility of Y (HREE) as observed in natural samples. Only in very fluorine-rich environments (e.g Pan and Fleet (1996); Harlov et al. (2006)) significant amounts of HREE are mobilized. This finding may indicate that the fluoride activity in the majority of metamorphic aqueous fluids is rather low. The reason for that could be the high affinity of fluorine to silicate (Dolejš and Zajacz, 2018; Dalou et al., 2015) which is one of the main components of aqueous phases in metamorphic
processes (Manning, 2018; Hermann et al., 2006).

As discussed by Ague (2017), the HREE mass change by fluid-rock interaction is much more determined by the mineral assemblages and phosphate mobility and therefore the halide content of the fluid phase might not be the only controlling factor for the HREE mobility. Nevertheless, the thermodynamic data reported here are consistent with the results of the HKF regression. Furthermore, the stability constants are affected by the formation of hydroxide mixed complexes and HCl/HF
formation during the thermodynamic integration. Therefore, the presented thermodynamic quantities can only be considered as semi-quantitative. Furthermore, it must be emphasized that the applied activity correction could be a source of huge uncertainty. As demonstrated by Hünenberger and McCammon (1999) does the Ewald summation, that is used to build up the periodic boundary condition within the simulation, disrupt the solvation free energy of highly charged ions. But this kind of perturbation is not accounted in the Debye-Hückel approach. Therefore, a more systematic evaluation of the impact of artificial periodic
electrostatics and neutralizing background charge on the computed thermodynamic properties derived from AIMD simulations especially at high temperatures is required in future studies.

*Author contributions.* J.S. and S.J. designed the study. J.S. performed the numerical simulations and S.J. supervised the analysis and interpretation of the results. Both authors discussed the results of the study and wrote the manuscript.

*Competing interests.* The authors declare no competing financial interest

*Acknowledgements.* This study was supported by the Deutsche Forschungsgemeinschaft in the framework of project JA 1469/10-1. The authors gratefully acknowledge the Gauss Centre for Supercomputing e.V. (www.gauss-centre.eu) for funding this project by providing computing time through the John von Neumann Institute for Computing (NIC) on the Supercomputer JUWELS and JURECA at Jülich Supercomputing Centre (JSC). We appreciate helpful discussions with Thomas Wagner.





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
