# Peer review of "Yttrium speciation in subduction zone fluids from *ab initio* molecular dynamics simulations"

_Solid Earth, 2019_

## Referee Comment (RC1) · Anonymous Referee #1 · 31 Jan 2020

This is an excellent manuscript on predicting the strengths of the complexes of Y with OH, F, and Cl. A great strength of the manuscript is that it takes pains to transform the results of the molecular simulations to thermodynamic equilibrium constants, so that the results can be compared with traditional thermodynamic estimation techniques.

This manuscript should be accepted with just minor corrections, as follows:

line 429....does not yield stable Ho/Y complexes... This statement seems to indicate that the authors think that a logK of less than zero is "not stable". It only shows that the complex is weak...a value of logK=0 has no special significance as it is a standard state quantity. The authors need to be more careful about the use of term "stable".

[Figure]

lines 450-451: again this confusing use of the word "stable". Just because the logK values are different doesn't mean F will predominate over Cl complexes. As the authors show in Fig. 12, the final concentration of a complex in a fluid depends also on the amount of available ligand. And that ligand might be taken up by much more abundant cations than $Y^{3+}$, e.g. $Mg^{2+}$ or $Ca^{2+}$, and so on.

lines 485-490: Also, $MgF^+$ and $CaF^+$ could take up all the $F^-$

---

## Referee Comment (RC2) · Yuan Mei (Referee) · 7 Feb 2020

Review of "Yttrium speciation in subduction zone fluids from ab initio molecular dynamics simulations" by Johannes Stefanski and Sandro Jahn

This manuscript presents a theoretical study of Y complexation using intensive ab initio MD simulations. The authors performed MD runs to describe the geometries and coordination of Y complexes, and used the thermodynamic integration method to calculate the free energy surface of Y-Cl/F reactions. These results have been used to calculate the thermodynamic properties at 800°C and high pressure. This study provides valuable knowledge of Y-Cl/F complexation under extreme high T-P conditions, which has significant implication in understanding Y and REE under conditions such as subduction zone. The computational methods are well documented and easy to follow. Overall it is a well-organised study, and the manuscript is very informative. It is worth to be published after revisions as noted.

Fig 2. Add constraint distance label for Fig2d.

Fig4: The x-axis is "AIMD run", which has no meanings… The authors could try to group runs with the same box to see the trend of changes.

I found the unconstraint MD part (section 3.1) is hard to follow. I understand it is challenge to put together large amount of data, especially for Y complexes whose geometry and coordinates are very disordered and complicated compare to some other elements.
The data listed in Table 1,3,4 are heavily overlapped, and it's hard to cross check those tables during reading. I suggest the authors consider to merge those Tables to one or two, and put relevant data in the one table.

One major information from the MD is the dissociation of $H^+$ during the simulation. As shown in Fig3, the difference of Y-O distances for $Y-OH^-$ and $Y-OH_2$ are very distinguishable. I suggest in Fig3, label the bond distances of each peak (or in the text); and in Table 3, show the $Y-OH^-$ and $Y-OH_2$ distances separately.

In practical, monitoring the change of Y-O distances helps us to identify the H dissociation during the dynamic run. For example, in Fig7, the authors can also add a dynamic distance plot to show the change of Y-O distance during the proton transfer.

The authors mentioned Na-Cl association in some MD runs. Please provide more details of the criteria of Na-Cl association (e.g., CN cutoff).

AIMD of run#5: The whole run looks like the "meta-stable" stage. As run#5 and #4 share the same box size and particles and just started with different initial configuration. The last formed structure of run #5 turned to be the first formed structure of run #4 (Table4). Can you discuss on this?

Line 235: "In #1, the highest amount of hydroxide is formed…". How about run #6? $Y-OH^-$ is 1 in run #6 (Table 3). It's not clear which runs# are discussed in this paragraph.

Fig5: This figure is very informative, but hard to read. A main confusion is the definition of "Y-OH formation" and "Occurrence of the initial complex". As when proton dissociated and Y-OH formed, it is not the "initial complex" anymore. The Y-Na formation could be

recognised as the 2$^{nd}$ shell interaction, but in the 1$^{st}$ shell, the number of hydration water changed in some calculations (e.g., #17), and the Y-OH$_2$ and Y-OH$^-$ are totally distinctive bond (as we can see from the bond distance), which shouldn't be classified as "Occurrence of the initial complex", unless change the definition to "initial Y-halide complex". Another suggestion is to move the legend to the top or bottom, so the figure can be larger in the published version.

Line 280-285: hydrated halide ions. What's the CN cutoff for Cl-/F- hydration? The hydration number of 4-5 at that density looks smaller compare to previous studies (e.g., Sherman 2007, Mei et al., 2018).

Fig7. Looks like the green curve are the running average of the constraint force. Can you show the dynamic force (e.g., Fig2(III)) to see how much difference is?

Line 319: "For the 4.5 GPa runs, the dissociation energies of the Y chloride complexes significantly decrease." Please specify the "significantly decrease". As shown in Table5, TI-1 is -36.1, TI-4 is -29.6, not significant decrease.

Line 325: "TI-5 yields the lowest dissociation energy of 8.5 kj mol $^{-1}$". I wouldn't say "lowest" here. As you didn't calculate the TI of reaction [YCl$_3$]$_{aq}$ = [YCl$_2$]$^+$ + Cl$^-$ , which would give lower FES.

Line 328: "For [YCl$_3$(H$_2$O)$_5$]$_{aq}$ it was not possible to derive a dissociation energy because the initial complex dissociated at short Y-Cl constraint distances within the first picosecond of each simulation." That's **incorrect**. It is possible to calculate the FES of this reaction, by restraint two of the Y-Cl at equilibrium bond distances (e.g., Fig1 of Mei et al., GCA, 179 (2016) 32-52). You would expect a low dissociation FES for that reaction as YCl$_3$ is not preferred complexes.

Line 335-340: "In the latter case at a constant distance of 2.6Å one of unconstrained fluorides separates from the initial complex. However, this behavior is not reproducible. " Again, you can restraint the Y-F pair to keep F around the equilibrium bond distance.

Fig8: For those chemical reactions, why using "→" in Fig8 but "=" in other tables? Please keep consistent.

Fig10: No Y-axes label in Fig10a.

---

## Author Comment (AC1) · 20 Mar 2020

We would like to thank the reviewer for useful comments. Here is our point by point reply (line numbers in the answers refer to the revised manuscript).

comment: line 429....does not yield stable Ho/Y complexes... This statement seems to indicate that the authors think that a logK of less than zero is "not stable". It only shows that the complex is weak...a value of logK=0 has no special significance as it is a standard state quantity. The authors need to be more careful about the use of term "stable".

[Figure]

answer: We thank the reviewer for making this valid point. We revised this notation in all paragraphs where a misleading interpretation could arise for the reader (see lines 8-10, 273, 293, 361, 382, 434, 456, 474). Stability is now used either in the context of mechanical stability during the MD simulation or of specific thermodynamic conditions.

comment: lines 450-451: again this confusing use of the word "stable". Just because the logK values are different doesn't mean F will predominate over Cl complexes. As the authors show in Fig. 12, the final concentration of a complex in a fluid depends also on the amount of available ligand. And that ligand might be taken up by much more abundant cations than $Y^{3+}$, e.g. $Mg^{2+}$ or $Ca^{2+}$, and so on.

answer: See answer to previous comment.

comment: lines 485-490: Also, $MgF^+$ and $CaF^+$ could take up all the $F^-$

answer: We further discuss the competition of different metal cations for $F^-$ in line 532.

---

## Author Comment (AC2) · 20 Mar 2020

Johannes Stefanski and Sandro Jahn

s.jahn@uni-koeln.de

We would like to thank Yuan Mei for her thorough review and useful comments. In the following we respond to all individual points (line numbers refer to the revised version of the manuscript).

C: Fig 2. Add constraint distance label for Fig2d.

A: We added the missing label.

C: Fig4: The x-axis is "AIMD run", which has no meanings... The authors could try to group runs with the same box to see the trend of changes.

[Figure]

A: As suggested we grouped the coordination numbers in Y-chloride, -fluoride and -mixed complexes.

C: I found the unconstraint MD part (section 3.1) is hard to follow. I understand it is challenge to put together large amount of data, especially for Y complexes whose geometry and coordinates are very disordered and complicated compare to some other elements. The data listed in Table 1,3,4 are heavily overlapped, and it's hard to cross check those tables during reading. I suggest the authors consider to merge those Tables to one or two, and put relevant data in the one table.

A: To improve the readability we merged Tables 3 and 4. Furthermore, we moved the list of formed complexes into the Supporting Information as well as the list of Y-H2O2nd distances.

C: One major information from the MD is the dissociation of H+ during the simulation. As shown in Fig3, the difference of Y-O distances for Y-OH– and Y-OH2 are very distinguishable. I suggest in Fig3, label the bond distances of each peak (or in the text); and in Table 3, show the Y-OH– and Y-OH2 distances separately.

A: We added the range of observed distances in lines 219-220. To not overload Table 3 we included the Y-OH- and Y-OH2 distances into Table SI1 in the Supplementary Information.

C: In practical, monitoring the change of Y-O distances helps us to identify the H dissociation during the dynamic run. For example, in Fig7, the authors can also add a dynamic distance plot to show the change of Y-O distance during the proton transfer.

A: Thank you for this suggestion. However, we think that adding a dynamic distance plot to Fig. 7 does not really add new information to what is already illustrated in the OH- plot (Fig. 7 III).

C: The authors mentioned Na-Cl association in some MD runs. Please provide more details of the criteria of Na-Cl association (e.g., CN cutoff).

A: We added further information of the NaCl association to Table SI1. The cutoff for the calculation of the Na-Cl coordination number was set to the first minimum of the gNaCl(r) in the range of 3.8 – 4.0 Å. A reference is added in line 223.

C:AIMD of run#5: The whole run looks like the "meta-stable" stage. As run#5 and #4 share the same box size and particles and just started with different initial configuration. The last formed structure of run #5 turned to be the first formed structure of run #4 (Table4). Can you discuss on this?

A: We included a statement of the meta-stable character of this simulations run in line 228.

C: Line 235: "In #1, the highest amount of hydroxide is formed...". How about run #6? Y-OH– is 1 in run #6 (Table 3). It's not clear which runs# are discussed in this paragraph.

A: The respective sentence was indeed confusing and we therefore deleted it.

C: Fig5: This figure is very informative, but hard to read. A main confusion is the definition of "Y-OH formation" and "Occurrence of the initial complex". As when proton dissociated and Y-OH formed, it is not the "initial complex" anymore. The Y-Na formation could be recognised as the 2nd shell interaction, but in the 1st shell, the number of hydration water changed in some calculations (e.g., #17), and the Y-OH2 and Y-OH– are totally distinctive bond (as we can see from the bond distance), which shouldn't be classified as "Occurrence of the initial complex", unless change the definition to "initial Y-halide complex". Another suggestion is to move the legend to the top or bottom, so the figure can be larger in the published version.

A: We thank you for your suggestions for improvement regarding Fig 5. We changed the misleading naming of the initial complex to "Occurrence of the initial Y-halide complex" and relocated the position of the legend.

C: Line 280-285: hydrated halide ions. What's the CN cutoff for Cl-/F- hydration? The

hydration number of 4-5 at that density looks smaller compare to previous studies (e.g., Sherman 2007, Mei et al., 2018).

A: Here we counted the hydrogen atoms bonded to $H_2O$ surrounding the chloride. The cutoff is taken from the first minimum of g(r)Cl-H at around 2.8 Å. Sherman (2007) and Mei et. al. (2018) integrated the g(r)Cl-O to derive the Cl-O coordination number. This leads to a higher number of $H_2O$ assigned to the first hydration shell of the ion due to overlapping coordination shells. Whereas the water molecules that form the first hydration shell of an anion are commonly characterized by their orientation where one of the hydrogens pointing towards the central anion (see Driesner 2013, Rev. Mineral. Geochem.).

In Fig. 4 of Mei et. al. (2018) the integration of g(r)Cl-H at 700°C and 6 GPa (60 kbar) with a density comparable to the high pressure runs of this study provides a Cl-H coordination of 6. Considering the temperature and pressure differences of 100 K and 1.5 GPa between the simulations the results of both studies are in good agreement.

C: Fig7. Looks like the green curve are the running average of the constraint force. Can you show the dynamic force (e.g., Fig2(III)) to see how much difference is?

A: We thank you for this useful advice. We added the dynamic force to Fig. 7. This further information improves the illustration.

C: Line 319: "For the 4.5 GPa runs, the dissociation energies of the Y chloride complexes significantly decrease." Please specify the "significantly decrease". As shown in Table5, TI-1 is -36.1, TI-4 is -29.6, not significant decrease.

A: This statement refers to the difference between TI-4 and TI-5. The sentence was revised.

C: Line 325: "TI-5 yields the lowest dissociation energy of 8.5 kj mol -1 ". I wouldn't say "lowest" here. As you didn't calculate the TI of reaction [YCl3]aq = [YCl2]+ + Cl- , which would give lower FES.

A: We have revised the statement.

C: Line 328: "For [YCl3(H2O)5]aq it was not possible to derive a dissociation energy because the initial complex dissociated at short Y-Cl constraint distances within the first picosecond of each simulation." That's incorrect. It is possible to calculate the FES of this reaction, by restraint two of the Y-Cl at equilibrium bond distances (e.g., Fig1 of Mei et al., GCA, 179 (2016) 32-52). You would expect a low dissociation FES for that reaction as YCl3 is not preferred complexes.

A: We added a short discussion of this approach and explained why we excluded this specific complex.

C: Line 335-340: "In the latter case at a constant distance of 2.6 ÅĿ one of uncon-strained fluorides separates from the initial complex. However, this behavior is not reproducible. " Again, you can restraint the Y-F pair to keep F around the equilibrium bond distance.

A: Because this was only observed for one integration step we have not applied more constraints.

C: Fig8: For those chemical reactions, why using "aÌĂ" in Fig8 but "=" in other tables? Please keep consistent.

A: The Figure was updated and the chemical equations are consistent now.

C: Fig10: No Y-axes label in Fig10a.

A: Thank you, we fixed this export error.

–––––––––––––––––––––––

---

## Author Response (AR1)

Dear Editor,

we would like to thank the two reviewers for carefully reading and commenting on our manuscript. We have revised the paper according to the very useful and constructive suggestions and questions. Below we provide a point-by-point response to the individual issues raised by the reviewers with reference to the line number, figure or table if changes in the manuscript were made.

Best regards,

Johannes Stefanski and Sandro Jahn
* * *
REVIEWER REPORT(S):

**Anonymous Referee #1**

line 429....does not yield stable Ho/Y complexes... This statement seems to indicate that the authors think that a logK of less than zero is "not stable". It only shows that the complex is weak...a value of logK=0 has no special significance as it is a standard state quantity. The authors need to be more careful about the use of term "stable".

We thank the reviewer for making this valid point. We revised this notation in all paragraphs where a misleading interpretation could arise for the reader (see lines 8-10, 273, 293, 361, 382, 434, 456, 474). Stability is now used either in the context of mechanical stability during the MD simulation or of specific thermodynamic conditions.

lines 450-451: again this confusing use of the word "stable". Just because the logK values are different doesn't mean F will predominate over Cl complexes. As the authors show in Fig. 12, the final concentration of a complex in a fluid depends also on the amount of available ligand. And that ligand might be taken up by much more abundant cations than Y3+, e.g. Mg2+ or Ca2+, and so on.

See previous comment.

lines 485-490: Also, MgF+ and CaF+ could take up all the F-

We further discuss the competition of different metal cations for $F^-$ in line 532.

**Yuan Mei (Referee)**

Fig 2. Add constraint distance label for Fig2d.

We added the missing label.

Fig4: The x-axis is "AIMD run", which has no meanings... The authors could try to group runs with the same box to see the trend of changes.

As suggested we grouped the coordination numbers in Y-chloride, -fluoride and -mixed complexes.

I found the unconstraint MD part (section 3.1) is hard to follow. I understand it is challenge to put together large amount of data, especially for Y complexes whose geometry and coordinates are very disordered and complicated compare to some other elements. The data listed in Table 1,3,4 are heavily overlapped, and it's hard to cross check those tables during reading. I suggest the authors consider to merge those Tables to one or two, and put relevant data in the one table.

To improve the readability we merged Tables 3 and 4. Furthermore, we moved the list of formed complexes into the Supporting Information as well as the list of Y-H$_2$O$^{2nd}$ distances.

One major information from the MD is the dissociation of H$^+$ during the simulation. As shown in Fig3, the difference of Y-O distances for Y-OH$^-$ and Y-OH$_2$ are very distinguishable. I suggest in Fig3, label the bond distances of each peak (or in the text); and in Table 3, show the Y-OH$^-$ and Y-OH$_2$ distances separately.

We added the range of observed distances in lines 219-220. To not overload Table 3 we included the Y-OH$^-$ and Y-OH$_2$ distances into Table SI1 in the Supplementary Information.

In practical, monitoring the change of Y-O distances helps us to identify the H dissociation during the dynamic run. For example, in Fig7, the authors can also add a dynamic distance plot to show the change of Y-O distance during the proton transfer.

Thank you for this suggestion. However, we think that adding a dynamic distance plot to Fig. 7 does not really add new information to what is already illustrated in the OH$^-$ plot (Fig. 7 III).

The authors mentioned Na-Cl association in some MD runs. Please provide more details of the criteria of Na-Cl association (e.g., CN cutoff).

We added further information of the NaCl association to Table SI1. The cutoff for the calculation of the Na-Cl coordination number was set to the first minimum of the g$_{NaCl}$(r) in the range of 3.8 – 4.0 Å. A reference is added in line 223.

AIMD of run#5: The whole run looks like the "meta-stable" stage. As run#5 and #4 share the same box size and particles and just started with different initial configuration. The last

formed structure of run #5 turned to be the first formed structure of run #4 (Table4). Can you discuss on this?

We included a statement of the meta-stable character of this simulations run in line 228.

Line 235: "In #1, the highest amount of hydroxide is formed...". How about run #6? Y-OH¯ is 1 in run #6 (Table 3). It's not clear which runs# are discussed in this paragraph.

The respective sentence was indeed confusing and we therefore deleted it.

Fig5: This figure is very informative, but hard to read. A main confusion is the definition of "Y-OH formation" and "Occurrence of the initial complex". As when proton dissociated and Y-OH formed, it is not the "initial complex" anymore. The Y-Na formation could be recognised as the 2$^{nd}$ shell interaction, but in the 1$^{st}$ shell, the number of hydration water changed in some calculations (e.g., #17), and the Y-OH$_2$ and Y-OH¯ are totally distinctive bond (as we can see from the bond distance), which shouldn't be classified as "Occurrence of the initial complex", unless change the definition to "initial Y-halide complex". Another suggestion is to move the legend to the top or bottom, so the figure can be larger in the published version.

We thank you for your suggestions for improvement regarding Fig 5. We changed the misleading naming of the initial complex to "Occurrence of the initial Y-halide complex" and relocated the position of the legend.

Line 280-285: hydrated halide ions. What's the CN cutoff for Cl-/F- hydration? The hydration number of 4-5 at that density looks smaller compare to previous studies (e.g., Sherman 2007, Mei et al., 2018).

Here we counted the hydrogen atoms bonded to H$_2$O surrounding the chloride. The cutoff is taken from the first minimum of g(r)$_{Cl-H}$ at around 2.8 Å. Sherman (2007) and Mei et. al. (2018) integrated the g(r)$_{Cl-O}$ to derive the Cl-O coordination number. This leads to a higher number of H$_2$O assigned to the first hydration shell of the ion due to overlapping coordination shells. Whereas the water molecules that form the first hydration shell of an anion are commonly characterized by their orientation where one of the hydrogens pointing towards the central anion (see Driesner 2013, Rev. Mineral. Geochem.).

In Fig. 4 of Mei et. al. (2018) the integration of g(r)$_{Cl-H}$ at 700°C and 6 GPa (60 kbar) with a density comparable to the high pressure runs of this study provides a Cl-H coordination of 6. Considering the temperature and pressure differences of 100 K and 1.5 GPa between the simulations the results of both studies are in good agreement.

Fig7. Looks like the green curve are the running average of the constraint force. Can you show the dynamic force (e.g., Fig2(III)) to see how much difference is?

We thank you for this useful advice. We added the dynamic force to Fig. 7. This further information improves the illustration.

Line 319: "For the 4.5 GPa runs, the dissociation energies of the Y chloride complexes significantly decrease." Please specify the "significantly decrease". As shown in Table5, TI-1 is -36.1, TI-4 is -29.6, not significant decrease.

This statement refers to the difference between TI-4 and TI-5. The sentence was revised.

Line 325: "TI-5 yields the lowest dissociation energy of 8.5 kj mol$^{-1}$ ". I wouldn't say "lowest" here. As you didn't calculate the TI of reaction $[YCl_3]_{aq} = [YCl_2]^+ + Cl^-$ , which would give lower FES.

We have revised the statement.

Line 328: "For $[YCl_3(H_2O)_5]_{aq}$ it was not possible to derive a dissociation energy because the initial complex dissociated at short Y-Cl constraint distances within the first picosecond of each simulation." That's **incorrect**. It is possible to calculate the FES of this reaction, by restraint two of the Y-Cl at equilibrium bond distances (e.g., Fig1 of Mei et al., GCA, 179 (2016) 32-52). You would expect a low dissociation FES for that reaction as $YCl_3$ is not preferred complexes.

We added a short discussion of this approach and explained why we excluded this specific complex.

Line 335-340: "In the latter case at a constant distance of 2.6 Å one of unconstrained fluorides separates from the initial complex. However, this behavior is not reproducible. " Again, you can restraint the Y-F pair to keep F around the equilibrium bond distance.

Because this was only observed for one integration step we have not applied more constraints.

Fig8: For those chemical reactions, why using "à" in Fig8 but "=" in other tables? Please keep consistent.

The Figure was updated and the chemical equations are consistent now.

Fig10: No Y-axes label in Fig10a.

We fixed this export error.

[revised manuscript text omitted]

Table SI1: List of atomic distances and coordination numbers. The cutoff distance to compute the Na-Cl/YCl-Na coordination numbers was set to 3.8-4.0 Å and to 3.0-3.5 Å for Na-F/YF-Na.

| Run ID | Cell | distances (Å) | | | coordination numbers | | | NaCl species distribution (in %) | | |
|---|---|---|---|---|---|---|---|---|---|---|
| | | $Y-OH_2$ | $Y-OH^-$ | $Y-O$ $(2^{nd})$ | $YCl-Na^\dagger$ | $YF-Na^\dagger$ | $Na-Cl^\star$ | $NaCl$ | $NaCl_2$ | $NaCl_3$ |
| #1 | A1 | 2.35 | 2.13 | 4.7 | 0.1 | 0.0 | 0.6 | 89 | 8 | 2 |
| #2 | A1 | 2.37 | 2.11 | 4.8 | 0.5 | 0.0 | 0.7 | 78 | 22 | 0 |
| #3 | A1 | 2.37 | - | 5 | 0.4 | 0.0 | 0.6 | 79 | 20 | 1 |
| #4 | A1 | 2.35 | - | 5.1 | 0.8 | 0.0 | 0.9 | 66 | 25 | 10 |
| #5 | A1 | 2.41 | - | 5.2 | 1.2 | 0.0 | 0.8 | 55 | 39 | 6 |
| #6 | A2 | 2.42 | 2.10 | 4.5 | 0.0 | 0.0 | 0.7 | 91 | 9 | 0 |
| #7 | A3 | 2.39 | - | 4.4 | 0.0 | 0.1 | 0.6 | 84 | 16 | 0 |
| #8 | A4 | 2.43 | - | 4.5 | 0.0 | 0.1 | 0.3 | 98 | 2 | 0 |
| #9 | A2 | 2.39 | 2.13 | 4.4 | 0.2 | 0.3 | 0.6 | 76 | 24 | 0 |
| #10 | A2 | 2.39 | 2.12 | 4.5 | 0.3 | 0.7 | 0.7 | 74 | 24 | 2 |
| #11 | A3 | 2.4 | - | 4.5 | 0.0 | 0.7 | 0.6 | 74 | 26 | 0 |
| #12 | A1 | 2.37 | 2.08 | 4.6 | 0.0 | 0.0 | 1.0 | 62 | 34 | 4 |
| #13 | B1 | 2.37 | 2.13 | 4.5 | 0.1 | 0.0 | 1.2 | 69 | 20 | 10 |
| #14 | B1 | 2.35 | 2.14 | 4.5 | 0.3 | 0.0 | 0.5 | 82 | 17 | 1 |
| #15 | B1 | 2.36 | 2.13 | 4.4 | 0.4 | 0.0 | 0.9 | 79 | 17 | 4 |
| #16 | B2 | 2.34 | 2.19 | 4.5 | 0.0 | 0.1 | 0.4 | 100 | 0 | 0 |
| #17 | B2 | 2.38 | 2.17 | 4.6 | 0.0 | 0.4 | 0.3 | 91 | 9 | 0 |
| #18 | B2 | 2.39 | 2.16 | 4.3 | 0.0 | 0.8 | 0.4 | 94 | 6 | 0 |
| #19 | B2 | 2.37 | 2.19 | 4.5 | 0.3 | 0.1 | 0.5 | 69 | 31 | 0 |
| #20 | B2 | 2.39 | 2.15 | 4.4 | 0.1 | 0.5 | 0.2 | 100 | 0 | 0 |
| #21 | B2 | 2.35 | 2.14 | 4.5 | 0.1 | 0.5 | 0.3 | 93 | 7 | 0 |
| #22 | B1 | 2.32 | 2.16 | 4.4 | 0.0 | 0.0 | 0.9 | 81 | 17 | 2 |

† Total number of sodium ions that are associated with the halide of the yttrium complex over the simulation run.

⋆ Mean coordination of sodium by chloride over the simulation run.

[revised manuscript text omitted]